# A FYVE zinc finger domain protein specifically links mRNA transport to endosome trafficking

Thomas Pohlmann[1], Sebastian Baumann[1†], Carl Haag[1], Mario Albrecht[2‡], Michael Feldbrügge[1*]

[1]Institute for Microbiology, Cluster of Excellence on Plant Sciences, Heinrich-Heine University Düsseldorf, Düsseldorf, Germany; [2]Max Planck Institute for Informatics, Saarbrücken, Germany

**Abstract** An emerging theme in cellular logistics is the close connection between mRNA and membrane trafficking. A prominent example is the microtubule-dependent transport of mRNAs and associated ribosomes on endosomes. This coordinated process is crucial for correct septin filamentation and efficient growth of polarised cells, such as fungal hyphae. Despite detailed knowledge on the key RNA-binding protein and the molecular motors involved, it is unclear how mRNAs are connected to membranes during transport. Here, we identify a novel factor containing a FYVE zinc finger domain for interaction with endosomal lipids and a new PAM2-like domain required for interaction with the MLLE domain of the key RNA-binding protein. Consistently, loss of this FYVE domain protein leads to specific defects in mRNA, ribosome, and septin transport without affecting general functions of endosomes or their movement. Hence, this is the first endosomal component specific for mRNP trafficking uncovering a new mechanism to couple mRNPs to endosomes.

*For correspondence: feldbrue@ hhu.de

Present address: †Cell and Developmental Biology, Centre for Genomic Regulation, Universitat Pompeu Fabra, Barcelona, Spain; ‡Institute for Knowledge Discovery, Graz University of Technology, Graz, Austria

Competing interests: The authors declare that no competing interests exist.

## Introduction

Trafficking of membranes is essential for intracellular logistics. Important membranous carriers are endosomes that transport lipids, proteins, and mRNAs. These large vesicular structures are well-known for their function in endocytosis, transporting plasma membrane proteins to their site of degradation in the lysosome/vacuole system (*Huotari and Helenius, 2011*; *Rusten et al., 2012*). However, they also carry out other functions, such as receptor recycling or cytoplasmic signalling, and are therefore considered to be multipurpose platforms (*Gould and Lippincott-Schwartz, 2009*). Early endosomes are characterised by the presence of Rab5-like small G proteins and their special lipid composition consisting of PI3P lipids (phosphatidylinositol 3-phosphate; *Stenmark et al., 2002*; *Kutateladze, 2006*). These lipids are recognised by distinct protein domains, such as the FYVE zinc finger (*Stenmark et al., 1996*).

Endosomes are actively transported along the microtubule cytoskeleton, which is particularly critical in highly polarised cells, such as neurons and fungal hyphae. In the latter, microtubule-dependent transport supports apical tip growth and secretion of hydrolytic enzymes. This process is streamlined for efficiency and defects in transport result in impaired polar growth and reduced fitness (*Peñalva et al., 2012*; *Riquelme and Sánchez-León, 2014*).

An emerging theme is the intimate linkage of membrane and mRNA trafficking during spatio-temporal control of gene expression (*Kraut-Cohen and Gerst, 2010*; *Jansen et al., 2014*). Important examples are the actin-dependent co-transport of mRNAs and ER (endoplasmic reticulum) during budding in *Saccharomyces cerevisiae* (*Schmid et al., 2006*) or the microtubule-dependent

**eLife digest** DNA contains the instructions to build proteins. These instructions are first copied to make a molecule of messenger RNA (or mRNA for short). A large machine called the ribosome then reads the mRNA molecule and translates it to build a protein.

Many proteins must get to particular locations in a cell to carry out their roles. For some proteins, this is achieved by transporting the mRNAs to the right location before they get translated, via a process called 'mRNA trafficking'. However, mRNAs do not move by themselves; instead they bind to a host of mRNA-binding proteins, and the ribosomes that are required for translation to take place. Cells also move proteins between different locations using small bubble-like structures called vesicles. These vesicles are surrounded by a membrane, and so this process is known as 'membrane trafficking'. Previous work has shown that these two processes are often linked, as vesicles can also carry mRNA molecules. But it is not fully understood how mRNA molecules are connected to vesicles.

Now, Pohlmann et al. have used a fungus called *Ustilago maydis* as a model system to investigate how mRNAs and vesicles can move together in cells that grow to form filament-like structures called hyphae. This fungus uses these filaments to penetrate into plant tissues and causes a disease called corn smut. The experiments revealed a vesicle protein called Upa1 that contains a new type of binding site that allows Upa1 to bring an important RNA-binding protein to the surface of vesicles. Since the RNA-binding protein binds mRNA and the translating ribosomes, this can explain how mRNAs can associate with membranes to move together along hyphae.

When Pohlmann et al. engineered fungi that lacked the gene for Upa1, these mutants had problems transporting their mRNAs and associated ribosomes. These findings reveal a direct connection between mRNA trafficking and membrane trafficking. Future studies could now investigate whether similar processes take place in other cells that grow as long filaments, such as plant pollen tubes or nerve cells. These studies might provide new insights into plant reproduction or brain activity.

co-transport of mRNAs and endosomes during hyphal growth (*Baumann et al., 2012*; *Göhre et al., 2013*). Key factors are RNA-binding proteins that recognise specific localisation sequences within target mRNAs. Together with accessory factors, such as the poly(A)-binding protein, they form large macromolecular complexes called mRNPs (messenger ribonucleoprotein particles, *Bullock, 2011*; *Eliscovich et al., 2013*; *Buxbaum et al., 2015*). At present, however, detailed mechanistic insights on the connection of mRNPs to membranes are scarce (*Jansen et al., 2014*).

The best fungal model system to study co-trafficking of endosomes and mRNAs is the corn pathogen *Ustilago maydis* (*Jansen et al., 2014*). Here, the switch from yeast-like to hyphal growth is essential for the infection of its host, and defects in this polar growth correlate with reduced fungal virulence (*Brefort et al., 2009*; *Vollmeister et al., 2012a*). In hyphae, endosomes shuttle extensively along the microtubule cytoskeleton throughout the entire length of the hyphae (*Steinberg, 2014*). Transport is mediated by a cytoplasmic dynein complex (*Straube et al., 2001*) transporting Rab5a-positive endosomes towards the microtubule minus-ends and the kinesin-3 type motor Kin3 transports in the opposite direction (*Schuster et al., 2011*). Since endosomes carry the SNARE Yup1 (soluble N-ethylmaleimide-sensitive-factor attachment receptor; *Wedlich-Söldner et al., 2000*) and are positive for Rab5a, they were classified as early endosomes, which have initially been proposed to mainly function in endocytosis and signalling (*Steinberg, 2012*; *Bielska et al., 2014*).

Recently, we discovered a novel function for these endosomes, namely mRNA transport throughout the hyphae (*Baumann et al., 2012*), a process that is critical for polar growth and unconventional secretion of the endochitinase Cts1 (*Becht et al., 2006*; *Koepke et al., 2011*). The key factor is the RNA-binding protein Rrm4 containing three N-terminal RRMs (RNA recognition motifs) for RNA-binding and two C-terminal PABC/MLLE domains (*Figure 1A*; *Becht et al., 2005*; *Zarnack and Feldbrügge, 2010*; *Baumann et al., 2012*; *Vollmeister et al., 2012b*). The latter is known from the cytoplasmic poly(A)-binding protein and functions as a binding pocket for peptides containing a PAM2 motif (PABP-interacting motif 2; *Albrecht and Lengauer, 2004*; *Kozlov et al., 2004*; *Jinek et al., 2010*; *Xie et al., 2014*).

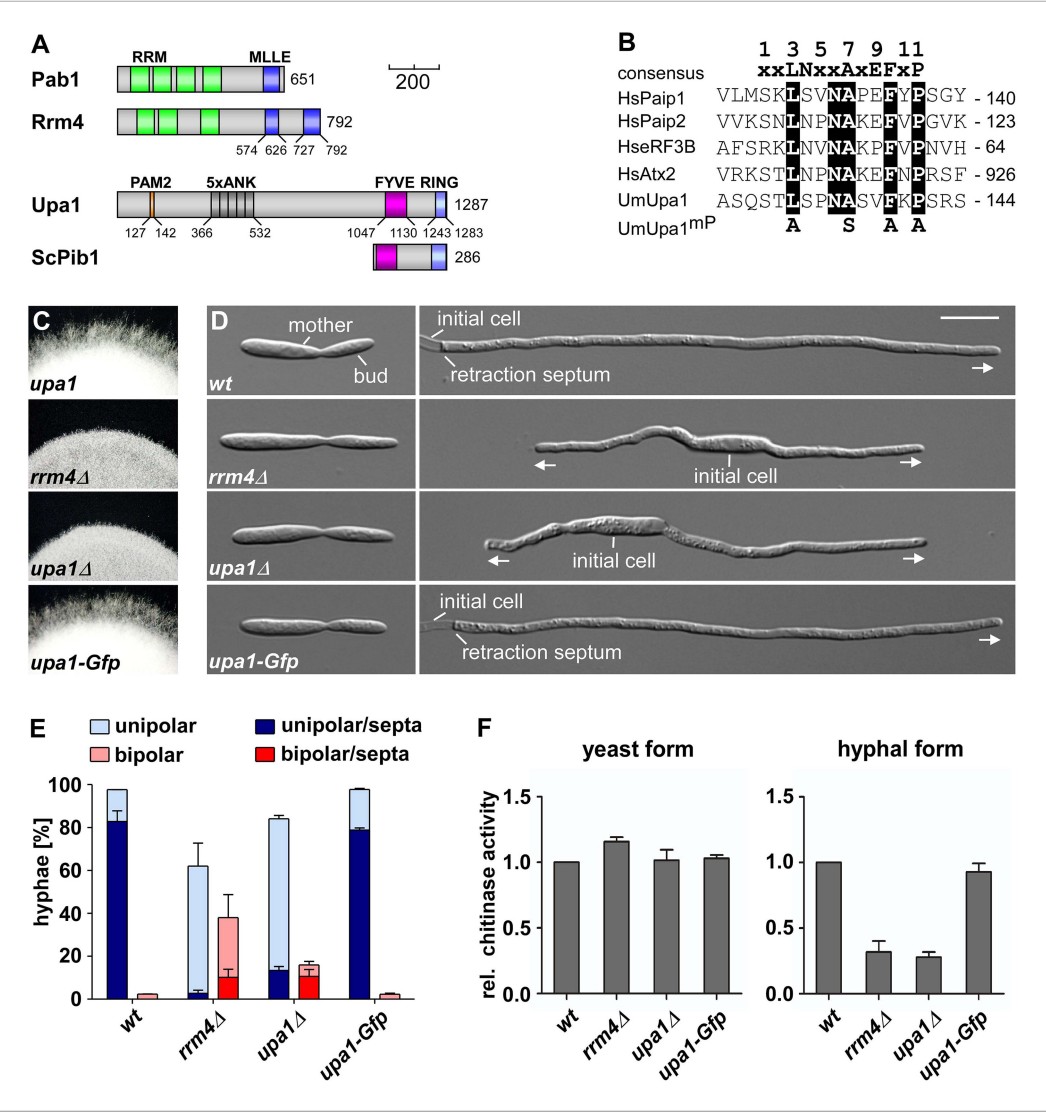

**Figure 1**. Loss of Upa1 causes defects in hyphal growth. (**A**) Schematic representation of proteins drawn to scale (bar, 200 amino acids) using the following colouring: green, RNA recognition motif (RRM); dark blue, MLLE domain (SMART E-values 6.7 and 0.35 for Rrm4, *Letunic et al., 2009*); red, PAM2 motif; dark grey, Ankyrin repeats; purple, FYVE domain; light blue, RING domain. (**B**) Comparison of PAM2 sequences found in Upa1 (accession number UMAG_12183) with those of human proteins, such as Paip1 (accession number NP_006442.2), Paip2 (accession number CAG38520.1), eRF3B (accession number CAB91089.1), and Atx2 (accession number NP_002964.3). (**C**) Edge of colonies growing on charcoal-containing medium under hyphae-inducing conditions (48 h.p.i.). (**D**) Growth of AB33 derivates in the yeast (left) and hyphal form (right; 8 h.p.i.; size bar, 10 μm). Growth direction is marked by arrows. (**E**) Percentage of hyphae (8 h.p.i.): unipolarity, bipolarity, and septum formation was quantified (error bars, s.e.m.; n = 3 independent experiments, >100 hyphae were counted per experiment; note that septum formation is given relative to the values of unipolar or bipolar hyphae set to 100%). (**F**) Relative chitinase activity mainly detecting endochitinase Cts1 (*Koepke et al., 2011*) in the yeast (left) or hyphal form (right; error bars, s.e.m.; n = 3 independent experiments).

Rrm4 specifically associates with shuttling Rab5a-positive endosomes (*Baumann et al., 2012*) and binds a specific set of mRNAs encoding, for example, the small G protein Rho3 or the septin Cdc3 (*König et al., 2009*). Studying Cdc3 in more detail revealed that not only its mRNA but also the protein is transported on endosomes in an Rrm4-dependent manner suggesting that endosome-coupled translation is crucial for septin localisation on these membranous carriers and needed for septin filamentation (*Baumann et al., 2014*). This was verified by demonstrating that translationally active ribosomes are transported on endosomes (*Higuchi et al., 2014*). Thus, Rrm4-dependent

transport carries out important general functions, such as distributing mRNAs (*König et al., 2009*; *Baumann et al., 2012*) and associated ribosomes (*Higuchi et al., 2014*), as well as more specific functions such as endosomal septin transport (*Baumann et al., 2014*). Despite the detailed knowledge on microtubule-dependent transport of endosome-coupled mRNA trafficking (*Jansen et al., 2014*), it is still unknown how mRNAs and associated proteins are connected to endosomes. Here, we identified a FYVE protein with specific functions in endosomal mRNP transport by coupling mRNPs to the shuttling vesicles.

## Results

### Upa1 is essential for efficient filamentous growth and secretion of Cts1

To identify factors connecting mRNPs to shuttling endosomes, we had two reasons for initially focusing on proteins containing the MLLE interaction motif PAM2 (*Albrecht and Lengauer, 2004*). First, two MLLE domain proteins, namely Rrm4 and Pab1 (*Figure 1A*), shuttle with Rab5a-positive endosomes along microtubules (*Baumann et al., 2012*), and second, mutations in the C-terminal MLLE domain of Rrm4 interfered with its movement (*Becht et al., 2006*). Closer inspection of Rrm4 revealed that it carries a second region with low similarity to the MLLE domain (*Figure 1A*, see below).

To find potentially interacting PAM2-containing proteins, we performed a HMM motif search for PAM2 screening the genome of *U. maydis* (*Albrecht and Lengauer, 2004*; *Kozlov et al., 2004*; *Kämper et al., 2006*). Among the 14 obtained candidates, *UMAG_12183* was particularly interesting because the encoded protein contained a lipid-binding FYVE domain, and its C-terminal domain architecture resembled the endosomal protein Pib1p from *S. cerevisiae* or mammalian Rififylin (*Figure 1A*; *Supplementary file 1*; *Shin et al., 2001*). In addition to the PAM2 motif (*Figure 1B*) and the FYVE domain, it contained five ankyrin repeats known to be protein–protein interaction interfaces (*Al-Khodor et al., 2010*), and a RING domain involved in ubiquitination (*Figure 1A*). The protein was designated Upa1 for the *U. maydis* PAM2 protein.

For functional analysis, we deleted *upa1* in the laboratory strain AB33 by homologous recombination (*Brachmann et al., 2004*). AB33 expresses an active heterodimeric transcription factor under control of the nitrate regulated *nar1* promoter. Since the active heterodimer is sufficient to elicit the morphological transition, hyphae can be induced synchronously and reproducibly by switching the nitrogen source (*Figure 1C–D*; *Brachmann et al., 2001*; *Baumann et al., 2012*).

Studying growth of *upa1Δ* cells revealed no mutant phenotype in the yeast form suggesting that cytokinesis including septa formation was not disturbed (*Figure 1D*, see below). However, loss of Upa1 caused defects in hyphal growth. At the colony level, shorter hyphae were observed, which were comparable to those of *rrm4Δ* strains (*Figure 1C*). At the cellular level, a significant proportion of *upa1Δ* cells grew bipolar in contrast to unipolar wild-type hyphae and those hyphae that grew unipolar inserted retraction septa with reduced frequency (*Figure 1D–E*). This again is reminiscent of the mutant phenotype of *rrm4Δ* strains (*Figure 1C–E*; *Baumann et al., 2014*). Since *rrm4Δ* mutants were also disturbed in unconventional secretion of Cts1, specifically during hyphal growth (*Koepke et al., 2011*; *Stock et al., 2012*), we determined extracellular chitinase activity. This revealed that Cts1 secretion was strongly reduced in the *upa1Δ* strain only in the hyphal form, which was comparable to *rrm4Δ* strains (*Figure 1F*). Thus, loss of Upa1 causes defects in hyphal growth and secretion of Cts1, two cellular processes that are regulated by Rrm4-mediated endosomal mRNA transport.

### Upa1 carries a functional PAM2 motif

The PAM2 motif is defined as an interaction interface of the MLLE domain of the poly(A)-binding protein (*Albrecht and Lengauer, 2004*). To address whether the predicted PAM2 motif in Upa1 (*Figure 1B*) interacts with the MLLE domain of Pab1 of *U. maydis*, we used the yeast two-hybrid assay. To this end, Pab1 (*König et al., 2009*) or Upa1 versions were fused at the N-terminus with the DNA-binding domain or activation domain of Gal4p, respectively (Matchmaker 3 system, Clontech). Constructs were transformed into the *S. cerevisiae* strain AH109, and control experiments were performed (*Figure 2—figure supplement 1*). Interaction was scored by growth on selection plates. Testing Pab1 with full length Upa1 revealed no interaction. However, assaying a version of Upa1 without the FYVE domain (Upa1$^{\Delta F}$) did show binding (*Figure 2A*), suggesting that the FYVE domain might have interfered with the nuclear localisation of the protein. Upa1$^{mP\Delta F}$ additionally carrying point mutations in the PAM2 motif (*Figure 2A*) or Upa1$^{\Delta N1\Delta F}$ with a deletion of the PAM2 region was

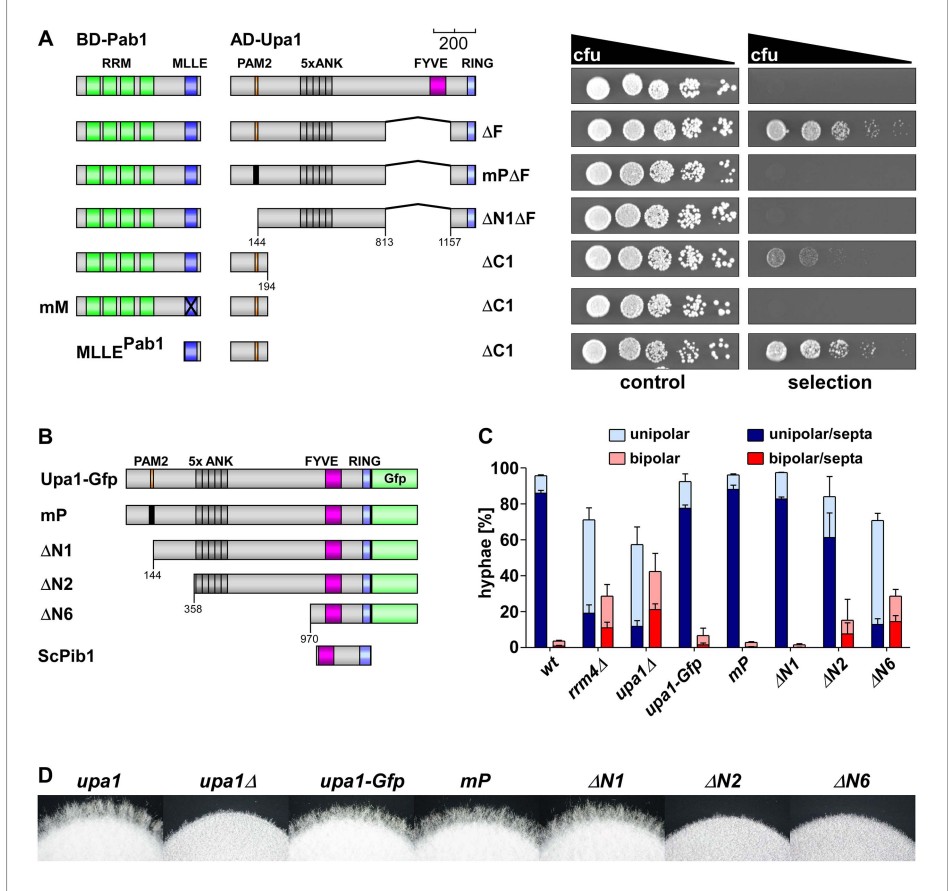

**Figure 2**. The PAM2 motif of Upa1 interacts specifically with the MLLE domain of Pab1. (**A**) Two-hybrid analysis with schematic representation of variants tested (left) and growth plates (right). Yeast cultures were serially diluted 1:5 (decreasing colony forming units, cfu) and spotted on respective growth plates assaying for reporter gene expression (see 'Materials and methods'). (**B**) Schematic representation of N-terminal truncated Upa1 variants fused at C-terminus with Gfp, drawn to scale (see **Figure 1A**; mP, mutation in the PAM2 motif indicated as black bar). (**C**) Percentage of hyphae (8 h.p.i.): unipolarity, bipolarity, and septum formation was quantified (error bars, s.e.m.; n = 3 independent experiments, >100 hyphae were counted for each strain per experiment; note that septum formation is given relative to the values of unipolar or bipolar hyphae set to 100%). (**D**) Edge of colonies growing on charcoal-containing medium under hyphae-inducing conditions (48 hr p.i).

The following figure supplements are available for figure 2:

**Figure supplement 1**. Interaction of Upa1 PAM2/Pab1 MLLE in vivo.

**Figure supplement 2**. The PAM2 motif is dispensable for Cts1 secretion.

no longer able to interact with Pab1, indicating that the PAM2 motif is necessary for binding (**Figure 2A**). Analysing only the first 194 amino acids (Upa1$^{\Delta C1}$) showed that this PAM2-containing region of Upa1 is sufficient for interaction (**Figure 2A**). Using a similar strategy for the interaction partner showed that the MLLE domain of Pab1 is necessary (Pab1$^{mM}$) and sufficient (MLLE$^{Pab1}$) for interaction with Upa1$^{\Delta C1}$ (**Figure 2A**). To verify these results in independent experiments, we demonstrated that an N-terminal part of Upa1 containing the PAM2 motif interacts with MLLE$^{Pab1}$ in GST pull down assays with the components expressed in *Escherichia coli* (see below). In summary, Upa1 contains a functional PAM2 motif that interacts with the MLLE of Pab1.

## The PAM2 motif is dispensable for the function of Upa1

In order to test the functional importance of the different Upa1 domains, we first generated a strain expressing Upa1 as functional C-terminal fusion protein with the enhanced version of the green

fluorescent protein (*Figure 1C–F*, Upa1-Gfp, eGFP). This was achieved by homologous recombination at the *upa1* locus of strain AB33 resulting in wild-type expression levels to avoid artefacts due to overexpression. Control experiments revealed that the amount of Upa1-Gfp did not change during the switch from yeast to hyphal growth (*Figure 2—figure supplement 2A*). To test PAM2 functionality, we generated Upa1 variants carrying a mutation in the PAM2 motif of Upa1 (Upa1$^{mP}$-Gfp) or a deletion of an N-terminal part containing that motif (Upa1$^{\Delta N1}$-Gfp; *Figure 2B*). The protein levels were comparable to that of wild type (*Figure 2—figure supplement 2B*). Testing for unipolar growth, for hyphal growth of colonies (*Figure 2C–D*), and for Cts1 secretion (*Figure 2—figure supplement 2C*) showed that the mutant strains did not differ from wild type. Hence, the Pab1 interacting motif PAM2 was dispensable for function.

To pinpoint functionally, important regions in the protein additional N-terminal deletions, Upa1$^{\Delta N2-6}$-Gfp (*Figure 2—figure supplement 2B*), were generated. Only expression of Upa1$^{\Delta N2}$-Gfp and Upa1$^{\Delta N6}$-Gfp was comparable to the wild-type level, and therefore, the function of Upa1$^{\Delta N3-5}$-Gfp could not be assessed (*Figure 2—figure supplement 2B*). Nevertheless, assaying Upa1$^{\Delta N2}$-Gfp revealed that the strain appeared to be slightly impaired in function (*Figure 2C–D*, *Figure 2—figure supplement 2C*), suggesting that there is a functionally important region at the N-terminus (143–357 aa). The Upa1$^{\Delta N6}$-Gfp expressing strain was indistinguishable from the *upa1Δ* strain, indicating a complete loss of function (*Figure 2C–D*; *Figure 2—figure supplement 2C*). Taken together, the PAM2 motif is dispensable for function of Upa1.

## Upa1 interacts with Rrm4 via two novel PAM2-like sequences

Next, we tested the interaction of Upa1 with the second MLLE domain-containing protein Rrm4 using the yeast two-hybrid assay (see above). Therefore, we fused Upa1-Gfp or Rrm4 versions to the DNA-binding domain or activation domain of Gal4p, respectively. Control experiments were performed as described above (*Figure 3—figure supplement 1A*). In contrast to Pab1, Rrm4 interacted with full length Upa1-Gfp (*Figure 3A*). Testing N-terminal truncations of Rrm4 revealed a minimal interaction domain containing the two predicted MLLE domains, while the C-terminal MLLE domain alone was not sufficient for binding (*Figure 3A*). Analysing C-terminal deletion of either this or a mutated MLLE domain demonstrated that in contrast to Pab1$^{MLLE}$, the domain is necessary but not sufficient for interaction with Upa1-Gfp (*Figure 3—figure supplement 1A*).

Screening N-terminal truncations of Upa1, we observed that, surprisingly, the PAM2 sequence was not needed for interaction with Rrm4 (Upa1$^{\Delta N1}$-Gfp, *Figure 3B*). Instead, the Rrm4-interacting region was mapped to the centre of Upa1 (*Figure 3B*, *Figure 3—figure supplement 1B,C*). Moreover and also unexpectedly, an Upa1 variant without the central Rrm4-interacting region was still able to interact with Rrm4 (Upa1$^{\Delta F}$-Gfp, *Figure 3B*), suggesting the presence of two yet unknown interacting regions. Therefore, we performed a more detailed analysis of the central interaction region of Upa1 (position 886 to 1030), applying linker scanning mutagenesis with 10 amino acid block mutations. Only a mutation in region 948 to 958 caused loss of binding (mutation 7 in *Figure 3C*). Interestingly, this region exhibited sequence similarity to the PAM2 motif (designated PAM2L, PAM2-like; *Figure 3D*), and furthermore, a second PAM2L sequence was found in the N-terminus of Upa1 (*Figure 3D*) supporting the initial two-hybrid data (*Figure 3B*). Mutating the conserved five amino acid core EFxxP or the highly conserved phenylalanine residue alone confirmed that these amino acids in both PAM2L motifs are crucial for the interaction with Rrm4 (*Figure 3—figure supplement 2*). A phylogenetic analysis of Upa1 revealed that both PAM2L sequences are conserved in related fungal proteins (*Figure 3E*, *Figure 3—figure supplement 3* and *Figure 3—figure supplement 4*). Thus, Upa1 interacts with Pab1 and Rrm4 via similar but not identical motifs, and a single PAM2L motif is sufficient for interaction.

To compare the binding specificities, we analysed the interaction of the Upa1 N-terminus containing PAM2 and the N-terminal PAM2L against full length Pab1 and Rrm4 as well as minimal regions of both proteins. Furthermore, we verified the observations performing GST co-purification experiments using protein variants expressed in *E. coli* (*Figure 3F*, *Figure 3—figure supplement 5* and *Figure 3—figure supplement 6*). The results were consistent with the mapping analysis (*Figure 3B–C*; *Figure 3—figure supplement 2*) showing that the MLLE-containing regions of Rrm4 and Pab1 exclusively recognise the corresponding PAM2L and PAM2 motifs, respectively. Importantly, we addressed whether the PAM2L motifs are also needed for Upa1 function during hyphal growth. To this end, we generated strains expressing Upa1-Gfp versions carrying

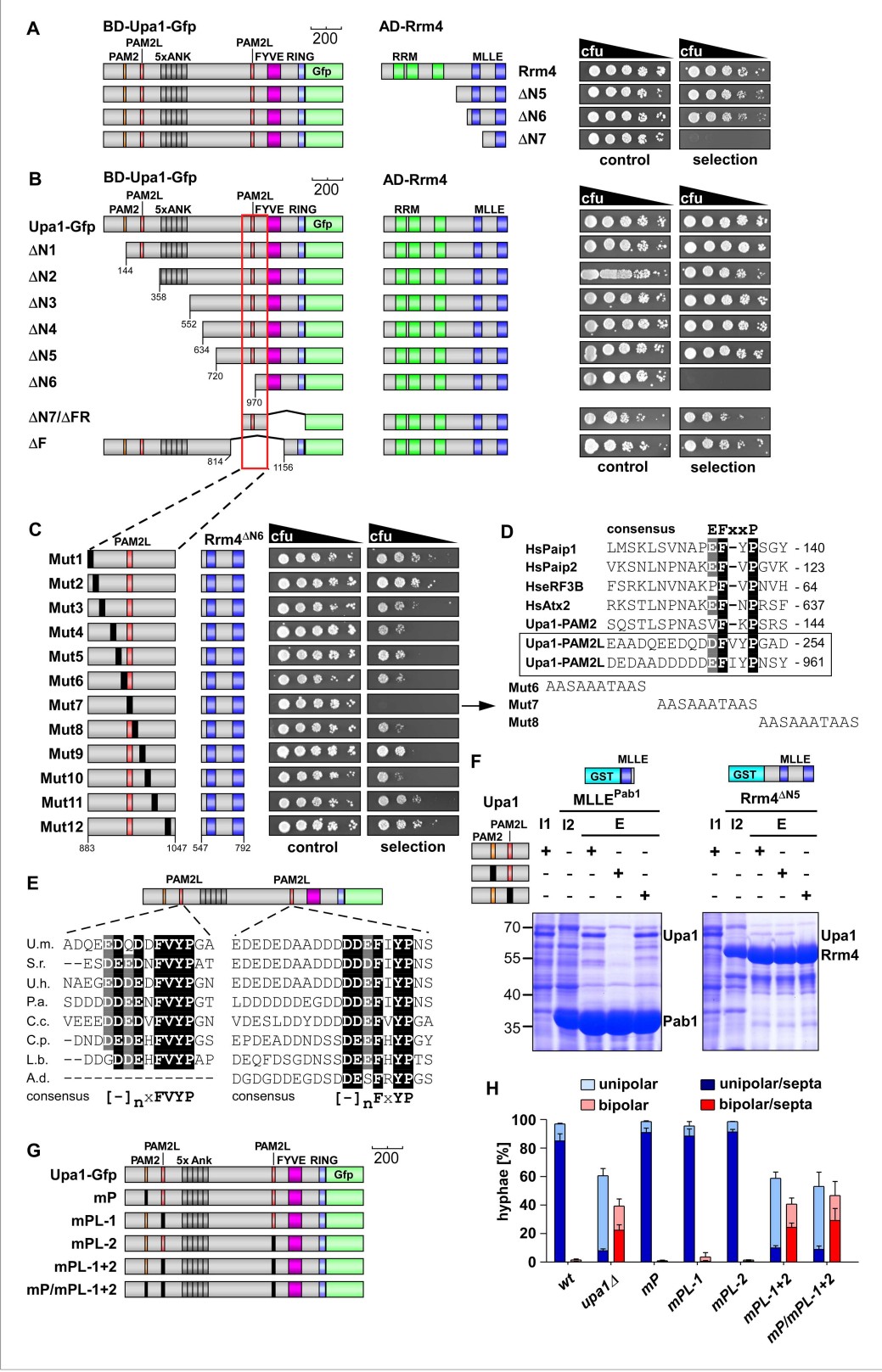

**Figure 3**. Upa1 contains two PAM2L motives for interaction with Rrm4. (**A**) Two-hybrid analysis with schematic representation of variants tested (left) and growth plates (right). Yeast cultures were serially diluted 1:5 (decreasing cfu) and spotted on respective growth plates assaying for reporter gene expression. (**B**) Two-hybrid analysis as in (**A**). Red rectangle indicates minimal region in Upa1 interacting with Rrm4. (**C**) Two-hybrid analysis as in (**A**). Upa1 region

*Figure 3. continued on next page*

*Figure 3. Continued*

identified in (**B**) was analysed by linker scanning mutagenesis (mutations indicated as black bar, Mut1-12). (**D**) Comparison of PAM2 and PAM2L sequences as in *Figure 1B*. Note, the second PAM2L motif was only mutated in Mut7 (**E**) PAM2L sequences of Upa1 compared to related sequences from basidiomycetes (U.m., *Ustilago maydis* UMAG_12183 / XP_758247.1; S.r., *Sporisorium reilianum*, accession number sr13323 / CBQ72642.1; U.h., *Ustilago hordei* accession number UHOR_03,485 / CCF52210.1; P.a. *Pseudozyma antarctica* GAK65366.1; C.c. *Coprinopsis cinerea* CC1G_00,427 / XP_001837291.2; C.p. *Coniophora putanea* XP_007767511.1; L.b. *Laccaria bicolor* XP_001876756.1; A.d. *Auricularia delicate* XP_007337909.1). (**F**) GST co-purification experiments with components expressed in *E. coli* N-terminal $His_6$-tagged versions of Upa1, Upa1$^{mP}$, and Upa1$^{mPL}$ (amino acids 1–363) were expressed to the same level (first input lane, I1; see *Figure 3—figure supplement 5B*). MLLE domains of Pab1 or Rrm4 (MLLE$^{Pab1}$ or Rrm4$^{\Delta N5}$, respectively) were expressed as GST fusion proteins (second input lane, I2). After GST affinity chromatography proteins were eluted (lanes marked with "E"). Interaction studies were performed with whole protein extracts from *E. coli* to demonstrate specific binding. (**G**) Schematic representation of Upa1 variants carrying mutations (black boxes) in the PAM2 and PAM2L regions. (**H**) Percentage of hyphae (8 h.p.i.): unipolarity, bipolarity, and septum formation was quantified (error bars, s.e.m.; n = 3 independent experiments, >100 hyphae were counted per experiment; note that septum formation is given relative to the values of unipolar or bipolar hyphae set to 100%).

The following figure supplements are available for figure 3:

**Figure supplement 1**. Upa1 interacts with Rrm4 in vivo.

**Figure supplement 2**. The evolutionarily conserved core of both PAM2L motifs is essential for interaction with Rrm4.

**Figure supplement 3**. Conserved PAM2L motif in the Upa1 N-terminal region.

**Figure supplement 4**. Conserved PAM2L motif in the central region of Upa1.

**Figure supplement 5**. Sequence specific recognition of the PAM2 and PAM2L sequence with the MLLE domains of Pab1 and Rrm4, respectively.

**Figure supplement 6**. Sequence specific recognition of the PAM2 and PAM2L sequence with the MLLE domains using purified components.

**Figure supplement 7**. The PAM2L motifs are functionally important for efficient secretion of Cts1.

---

mutations in the PAM2L motifs (*Figure 3G*). Scoring unipolar growth and secretion of Cts1 revealed consistent results leading to the conclusion that one PAM2L motif is sufficient for function, but if both are mutated functionality of Upa1 is lost. Additional mutations in the PAM2 motif made no difference indicating that it is the PAM2L motifs that are indeed crucial for activity (*Figure 3G–H*; *Figure 3—figure supplement 7*). In essence, Upa1 interacts directly with Rrm4 via two novel, functionally important PAM2L motifs.

## Upa1 shuttles on endosomes along microtubules

For further support of these interaction studies, we investigated the subcellular localisation of Upa1-Gfp in hyphae of *U. maydis*. The protein localised exclusively in the cytoplasm and was mainly present on distinct units that shuttled bidirectionally throughout the hyphae (*Figure 4A–B*; *Video 1*). No staining of other specific compartments, such as vacuoles was visible (*Figure 4A*). The observed motility was comparable to the bidirectional movement of Rrm4-Gfp and Pab1-Gfp (*Figure 4B*; *Figure 4—figure supplement 1A–C*; *Videos 2,3*) that are known to shuttle on Rab5a-positive endosomes (see below; *Baumann et al., 2012*; *Baumann et al. 2014*). Note that in contrast to Pab1-Gfp, the cytoplasmic signal of Upa1-Gfp is weak suggesting that the Upa1/Pab1 interaction is restricted to shuttling units (*Figure 4—figure supplement 1A–C*).

Upa1-Gfp movement was inhibited by treatment with the microtubule inhibitor benomyl (*Figure 4C*; *Video 4*) and deletion of *kin3*, which encodes the plus-end directed motor for endosomal movement, resulted in the accumulation of Upa1-Gfp signals in the centre of the cells

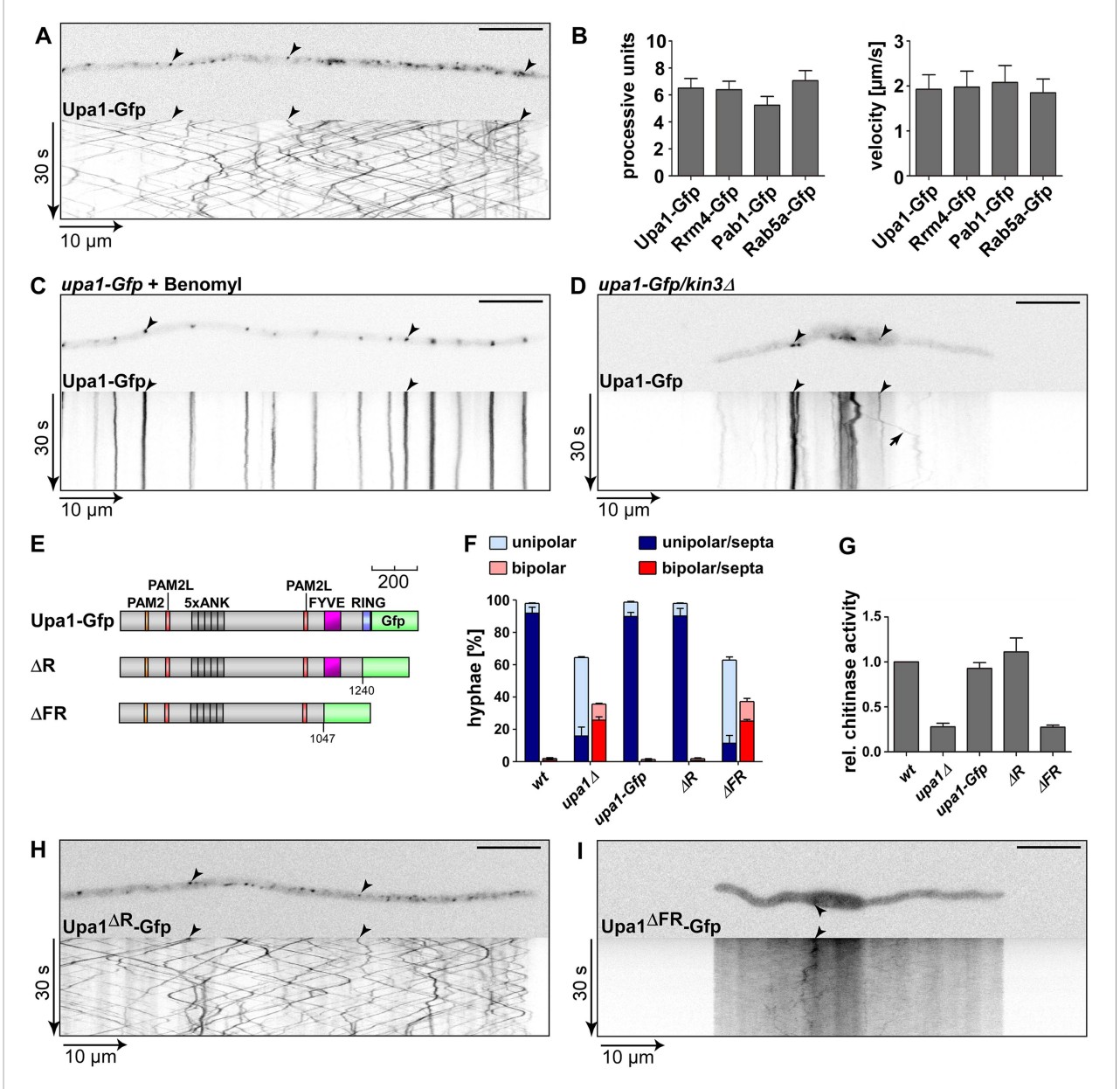

**Figure 4**. Endosomal targeting of Upa1 is functionally important. (**A**) Micrograph (size bar, 10 μm) and corresponding kymograph of hyphae expressing Upa1-Gfp showing bidirectional movement of signals as diagonal lines (arrowheads, *Video 1*). (**B**) Bar diagrams depicting amount of processive Upa1-Gfp signals (left, processive units per 10 μm hyphal length to accommodate for size differences between individual hyphae; error bars, s.d.; more than 30 hyphae per strain) and their velocity (right; velocity of tracks showing >5 μm processive movement; error bars, s.d.; 10 to 12 hyphae and more than 900 tracks per strain). (**C**) Hyphae treated with microtubule inhibitor benomyl. Micrograph (size bar, 10 μm) and corresponding kymograph showing static signals as vertical lines (arrowheads; *Video 4*). (**D**) Hyphae expressing Upa1-Gfp and carrying deletion in *kin3*. Micrograph (size bar, 10 μm) and corresponding kymograph showing static signals as vertical lines (arrowheads; *Video 5*). Arrow points towards residual movement. (**E**) Schematic representation of Upa1 fused at C-terminus with Gfp drawn to scale (see *Figure 1A*). (**F**) Percentage of hyphae (8 h.p.i.): unipolarity, bipolarity, and septum formation was quantified (error bars, s.e.m.; n = 3 independent experiments, >100 hyphae were counted per experiment; note that septum formation is given relative to the values of unipolar or bipolar hyphae set to 100%). (**G**) Relative chitinase activity mainly detecting endochitinase Cts1 in the hyphal form (*Koepke et al., 2011*; error bars, s.e.m., n = 3 independent experiments). (**H, I**) Micrographs (size bar, 10 μm) and corresponding kymographs of hyphae expressing Upa1^ΔR-Gfp (**H**) or Upa1^ΔFR-Gfp (**I**) (Videos 10,11).

The following figure supplement is available for figure 4:

**Figure supplement 1**. The FYVE domain is crucial for function.

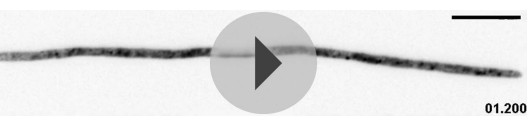

**Video 1.** Upa1-Gfp moves bidirectionally in a hypha of AB33upa1-Gfp. Video corresponds to *Figure 4A* (size bar = 10 µm, timescale in seconds, 200 ms exposure time, 150 frames, 5 frames/s display rate; QuickTime format, 6275 kB).

**Video 2.** Rrm4-Gfp moving bidirectionally in a hypha of AB33rrm4-Gfp. Video corresponds to *Figure 4—supplement figure 1B* (size bar = 10 µm, timescale in seconds, 150 ms exposure time, 150 frames, 6 frames/s display rate; QuickTime format, 1536 kB).

**Video 3.** Pab1-Gfp moving bidirectionally in a hypha of AB33pab1-Gfp. Note, that in contrast to Upa1-Gfp and Rrm4-Gfp (*Videos 1* and *2*) a higher background signal can be attributed to non-transported poly-adenylated mRNAs. Video corresponds to *Figure 4—supplement figure 1C* (size bar = 10 µm, timescale in seconds, 150 ms exposure time, 150 frames, 6 frames/s display rate; QuickTime format, 473 kB).

where minus-ends of microtubules are located (*Figure 4D*; *Video 5*; *Baumann et al., 2012*). Hence, Upa1 appears to localise on endosomes that shuttle along microtubules.

To map the endosome binding domain, we analysed the subcellular localisation of the respective mutated versions Upa1$^{mP}$-Gfp, Upa1$^{\Delta N1}$-Gfp, -$^{\Delta N2}$ and -$^{\Delta N6}$ (*Figure 3A*). This revealed that all versions containing the FYVE domain shuttled on endosomes (*Figure 4—figure supplement 1G*; *Videos 6–9*) consistent with the assumption that the FYVE domain is sufficient for endosome interaction.

For functional analysis of the Upa1 C-terminus containing the RING and FYVE domain, we expressed corresponding C-terminal truncations (*Figure 4E*, *Figure 4—figure supplement 1D-F*). Phenotypic analysis revealed that the RING domain (Upa1$^{\Delta R}$-Gfp) was dispensable for function under the tested conditions, whereas deletion of a C-terminal region containing the RING and FYVE domain (Upa1$^{\Delta FR}$Gfp) results in loss of function (*Figure 4F–G*, *Figure 4—figure supplement 1E*). Upa1$^{\Delta R}$-Gfp still shuttled on endosomes, but the additional deletion of the FYVE domain in Upa1$^{\Delta FR}$Gfp abolished movement (*Figure 4H–I*; *Videos 10,11*). In summary, the FYVE domain of Upa1 mediates endosomal localisation and, importantly, endosomal targeting of Upa1 is crucial for its function during polar growth and Cts1 secretion.

## Upa1 is specifically needed for the mRNP transport function of endosomes

To study whether Upa1 is indeed part of the endosomal compartment that is positive for Rab5a and Rrm4, Upa1-Gfp was expressed first with Rab5a-Cherry, an N-terminal fusion of Rab5a with the red fluorescent protein mCherry (*Baumann et al., 2012*). Rab5a-Cherry localises to shuttling endosomes and exhibits additional staining in the cytoplasm (*Figure 5A*; *Video 12*), which most likely marks the endomembrane system proposed to be late endosomes involved in endocytosis (*Higuchi et al., 2014*). Dynamic co-localisation experiments using dual view technology and msALEX microscopy (millisecond alternating laser excitation, *Baumann et al., 2014*) revealed that Upa1-Gfp co-localises extensively with motile Rab5a-positive endosomes (*Figure 5A*, *Figure 5—figure supplement 1A*; about 90% percent in both directions). A second marker for this motile endosomal compartment is Rrm4, which in contrast to Rab5a does not stain other membrane compartments. Consistently, the vast majority of Upa1-Gfp signals co-localises with Rrm4-Rfp (*Figure 5B*, *Figure 5—figure supplement 1B*; *Video 13*, about 90% percent in both directions). Thus, Upa1-Gfp is present on almost all Rab5a- and Rrm4-positive endosomes.

Next, we analysed the influence of the loss of Upa1 on the multiple functions of Rab5a-positive endosomes. Previously, it was shown that these endosomes function in cytokinesis and cell separation. For example, loss of the FYVE domain-containing guanine nucleotide exchange factor (GEF) Don1 or the plus-end directed Kinesin-3 type motor Kin3 resulted in the formation of cell aggregates (*Schink and Bölker, 2009*). Closer inspection revealed no cell separation defect in *upa1Δ* or *rrm4Δ* strains

**Video 4.** Upa1-Gfp movement is dependent on the microtubule cytoskeleton. Treating AB33upa1-Gfp for one hour with 50 µM microtubule destabilising drug benomyl, inhibits the bidirectional movement of Upa1-Gfp. Video corresponds to *Figure 4C* (size bar = 10 µm, timescale in seconds, 200 ms exposure time, 150 frames, 5 frames/s display rate; QuickTime format, 2862 kB).

**Video 5.** Loss of the plus-end directed kinesin-3 Kin3 disturbs Upa1-Gfp movement in AB33upa1-Gfp/kin3Δ. Immobile Upa1-Gfp accumulations can be seen in the middle of the cell. Residual movement might be due to dynein activity. Video corresponds to *Figure 4D* (size bar = 10 µm, timescale in seconds, 200 ms exposure time, 150 frames, 5 frames/s display rate; QuickTime format, 2382 kB).

(*Figure 2B*; *Figure 5—figure supplement 1C,D*) indicating that Upa1, like Rrm4 (*Becht et al., 2005*; *Baumann et al., 2014*), is not involved in endosomal functions during cytokinesis.

Another function for these Rab5a-positive endosomes is their role in endocytosis. This is mainly based on the observation that the styryl dye FM4-64 follows the endocytotic pathway by initially staining the plasma membrane followed by staining Rab5a-positive shuttling endosomes and lastly vacuoles (*Higuchi et al., 2014*). Testing endocytotic uptake of FM4-64 revealed no differences in the uptake and labelling of shuttling endosomes when comparing wild-type and *upa1Δ* strains (*Figure 5—figure supplement 1E*), suggesting that Upa1 is not involved in endocytosis. Next, we tested the shuttling of Rab5a-Gfp in hyphae comparing wild-type and *upa1Δ* strains. This showed neither a difference in the velocity of Rab5a-Gfp-positive endosomes nor in the bidirectional movement of Rab5a (*Figure 5C–D*, *Figure 5H*, *Figure 5—figure supplement 1F*; *Videos 14,15*). Hence, the endosomal protein Upa1 is not essential for general endosome functions.

However, testing the impact on Rrm4-Gfp revealed that its movement was drastically impaired in *upa1Δ* strains. Although the velocity of processive Rrm4-Gfp signals is the same as in wild type (*Figure 5—figure supplement 1F*), in the absence of Upa1, we observed fewer processive signals and a significant increase in signals exhibiting corralled movement (*Figure 5E–G*, *Figure 5—figure supplement 2B*; *Videos 16,17*). This altered movement was particularly eminent when quantifying processive signals reaching the apical region of hyphal tips (*Figure 5H*). This suggests that endosome association of Rrm4 was impaired. Analysing the altered movement of Rrm4 revealed that it was still microtubule-dependent (*Figure 5—figure supplement 3A*). The decline in processive Rrm4-Gfp signals was not due to a reduced protein amount, since expression is comparable in wild-type and *upa1Δ* strains (*Figure 5—figure supplement 3B*). Importantly, the processive Rrm4-Gfp signals co-localised with Rab5a and the endosomal marker protein Yup1, as well as with the lipophilic dye FM4-64. These co-localisation studies confirm that Rrm4 movement was specifically altered in the absence of Upa1 and that residual processive movement took place on endosomes (*Figure 5I–J*; *Figure 5—figure supplement 4*). Rrm4-Gfp signals exhibiting corralled movement did not co-localise with the membrane markers used, suggesting that these large accumulations constitute aberrant forms that fail to associate with the machinery for long-distance transport. Furthermore, addressing the endosomal shuttling of Upa1-Gfp in the absence of Rrm4 revealed that Upa1-Gfp movement is indistinguishable from wild type, suggesting that Upa1 attaches to endosomes independently of Rrm4 (*Figure 5—figure supplement 3C,D*). In summary, Upa1 is dispensable for general endosomal functions but is crucial for endosomal recruitment of Rrm4.

## Upa1 functions in endosomal targeting and transport of mRNAs, as well as associated ribosomes

Previously, it was shown that Rrm4 functions in endosomal transport of mRNAs and associated ribosomes (*Baumann et al., 2012*, *2014*; *Higuchi et al., 2014*). To address whether these functions are altered in the absence of Upa1, we first studied the movement of Pab1-Gfp, which is an established marker for mRNA (*Baumann et al., 2012*, *2014*). In contrast to wild type, the number of Pab1-Gfp-positive, processive signals reaching the apical pole was strongly reduced in the

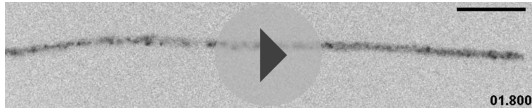

**Video 6.** Upa1^mP^-Gfp moving bidirectionally in a hypha of AB33upa1^mP^-Gfp. Mutating the PAM2 motif by amino acid exchanges does not inhibit movement of Upa1^mP^-Gfp. Video corresponds to *Figure 4—supplement figure 1G* top left (size bar = 10 μm, timescale in seconds, 200 ms exposure time, 150 frames, 5 frames/s display rate; QuickTime format, 2901 kB).

**Video 7.** Upa1^ΔN1^-Gfp moving bidirectionally in a hypha of AB33upa1^ΔN1^-Gfp. Loss of the first 143 amino acids including the PAM2 motif does not inhibit movement of Upa1^ΔN1^-Gfp. Video corresponds to *Figure 4—supplement figure 1G* top right (size bar = 10 μm, timescale in seconds, 200 ms exposure time, 150 frames, 5 frames/s display rate; QuickTime format, 2695 kB).

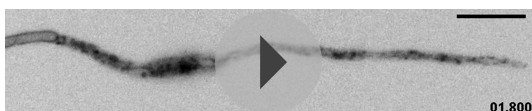

**Video 8.** Upa1^ΔN2^-Gfp moving bidirectionally in a hypha of AB33upa1^ΔN2^-Gfp. Loss of the first 337 amino acids including the PAM2 motif does not inhibit movement of Upa1^ΔN2^-Gfp although an increase in stationary background signals can be seen. Video corresponds to *Figure 4—supplement figure 1G* bottom left (size bar = 10 μm, timescale in seconds, 200 ms exposure time, 150 frames, 5 frames/s display rate; QuickTime format, 5614 kB).

**Video 9.** Upa1^ΔN6^-Gfp moving bidirectionally in a hypha of AB33upa1^ΔN6^-Gfp. The C-terminal part of Upa1 containing the FYVE and RING domains is sufficient for movement although an increase in stationary background signals can be seen. Video corresponds to *Figure 4—supplement figure 1G* bottom right (size bar = 10 μm, timescale in seconds, 200 ms exposure time, 150 frames, 5 frames/s display rate; QuickTime format, 4388 kB).

*upa1Δ* strain, even though the protein level in both strains are comparable (*Figure 6A–C*, *Figure 5—figure supplement 2C*; *Videos 18,19*). The processive signals exhibited the same velocity in wild type and *upa1Δ* cells (*Figure 5—figure supplement 1F*) indicating that the transport machinery itself is not affected whereas the loading of the mRNA cargo to endosomes is (*Figure 6B–C*). Consistently, the number of signals exhibiting corralled movement increased, and the number of signals reaching the apical pole decreased as described for Rrm4 (*Figure 6C*, *Figure 5—figure supplement 2*). These results suggest that the disturbed Rrm4 localisation in the absence of Upa1 also affects mRNA loading onto endosomes.

Next, we studied endosomal transport of Gfp-labelled ribosomal proteins of the small and large subunit in dependence of Upa1. Endosomal movement of ribosomal proteins was visualised by bleaching an area about 10 μm from the hyphal tip followed by video microscopy (*Baumann et al., 2014*, *2015*; *Figure 6D–G*). Rps2-Gfp, Rpl25-Gfp, and Rps19-Gfp exhibited bidirectional movement in shuttling units (*Figure 6E–G*), which were previously shown to be Rrm4-positive endosomes (*Baumann et al., 2014*; *Higuchi et al., 2014*). Note, that whereas Rps2-Gfp and Rpl25-Gfp were expressed ectopically, Rps19-Gfp was expressed at the homologous locus under control of the endogenous promoter resulting in stronger signals. Loss of Upa1 caused a severe reduction in shuttling signals (analysing five hyphae each revealed 42, 47, and 78 processive signals in Rps2-Gfp, Rpl25-Gfp, and Rps19-Gfp in wild type but 0, 1, 5 in *upa1Δ* strains, respectively). Thus, in the absence of Upa1, Rrm4 functions such as transport of mRNAs and associated ribosomes are disturbed.

## Upa1 is crucial for Rrm4-dependent endosomal transport of septin *cdc3* mRNA and Cdc3 protein

To test whether Rrm4-dependent septin mRNA transport was also affected (*Baumann et al., 2014*), we used λN-based RNA live imaging by expressing a *cdc3* mRNA with 16 BoxB in its 3′ UTR,

**Video 10.** Upa1$^{\Delta R}$-Gfp moving bidirectionally in a hypha of AB33upa1$^{\Delta R}$-Gfp. The C-terminal part comprising aa 1241–1287 including the RING domain is not necessary for shuttling of Upa1. Video corresponds to *Figure 4H* (size bar = 10 μm, timescale in seconds, 200 ms exposure time, 150 frames, 5 frames/s display rate; QuickTime format, 4069 kB).

**Video 11.** Upa1$^{\Delta FR}$-Gfp shows an even distribution in a bipolar growing hypha of AB33upa1$^{\Delta FR}$-Gfp. Thus, the C-terminal part comprising of aa 1048–1287 including the FYVE and RING domains is necessary for localising Upa1 to endosomes. Video corresponds to *Figure 4I* (size bar = 10 μm, timescale in seconds, 200 ms exposure time, 150 frames, 5 frames/s display rate; QuickTime format, 5201 kB).

and the λN RNA-binding peptide fused with a nuclear localisation signal (NLS) and triple Gfp (λN*$^{NLS}$-Gfp$^3$, *Figure 7A*; *Baumann et al., 2014*, *2015*). In this experimental set-up, unbound λN*$^{NLS}$-Gfp$^3$ was redirected to the nucleus, improving the cytoplasmic background signal (*Figure 7—figure supplement 1A*; *Video 20*). Consistent with earlier results, λN*$^{NLS}$-Gfp$^3$-labelled *cdc3B*$^{16}$ mRNA co-localised with Rrm4-Cherry-positive endosomes (*Figure 7B*; *Baumann et al., 2014*). However, deletion of *upa1* resulted in impaired *cdc3B*$^{16}$ mRNA transport (*Figure 7C–D*). We observed fewer processive particles and those that were detected showed a lower range of movement (*Figure 7C–D*). For the few examples that exhibited processive movement over a certain distance, the velocity was comparable to wild type (*Figure 7D*) indicating that the mRNAs can move with the speed of endosomes, but the attachment appeared to be less stable.

Rrm4 is needed for the correct localisation of Cdc3 protein on endosomes and in septin filaments forming a gradient that emanates from the hyphal tip (*Baumann et al., 2014*). Analysing the subcellular localisation of functional Cdc3-Gfp revealed that its localisation on endosomes was severely disturbed in *upa1Δ* strains (*Figure 7—figure supplement 1B*). Thus, without Upa1, hardly any Cdc3 protein could be detected on endosomes. Also the subcellular localisation of Cdc3-Gfp in septin filaments was altered in *upa1Δ* strains. Similar to *rrm4Δ* strains, septin filaments were still formed, but the gradient at the hyphal tip was lost (*Figure 7E*). To verify that the disturbed formation of septin filaments correlated with altered endosomal delivery of septin protein in the absence of Upa1, we performed fluorescence recovery after photobleaching (FRAP) experiments to analyse hyphal tips (*Baumann et al., 2014*). Due to the long maturation time of Gfp in the order of several minutes, local translation of newly synthesised protein at the hyphal tip can be excluded *Baumann et al., 2014*). Using the identical set-up as described before (*Baumann et al., 2014*), we determined a half time of recovery ($t_{1/2}$) of 4.2 min for wild-type hyphae (*Figure 7F*). In *upa1Δ* hyphae, $t_{1/2}$ was substantially increased to 14 min (*Figure 7F*, *Baumann et al., 2014*) confirming that Upa1-dependent septin transport is crucial for efficient assembly into filaments at the hyphal tip. In essence, these results demonstrate consistently, that Upa1 is of specific importance for Rrm4-dependent endosome functions.

## Discussion

### A novel FYVE domain protein containing PAM2 and PAM2L motifs for interaction with different MLLE proteins

Aiming at the identification of endosomal components involved in mRNP transport, the PAM2 protein Upa1 caught our attention because of its FYVE and RING domains. This domain organisation is similar to Pib1p in *S. cerevisiae* and mammalian Rififylin, two proteins which appear to function in endosomal protein sorting. Although their precise roles are still unclear (*Shin et al., 2001*; *Coumailleau et al., 2004*), they might function in ubiquitination during protein sorting due to the presence of the RING domain found in RNF-type E3 ubiquitin ligases (*Nikko and Pelham, 2009*).

The FYVE domain interacts with PI$_3$P lipids and thereby targets proteins to endosomes and endocytotic vesicles (*Stenmark et al., 2002*; *Lee et al., 2005*; *Kutateladze, 2006*). Consistently, it was already demonstrated in *U. maydis* that the guanine GEF Don1 is targeted to Rab5a-positive

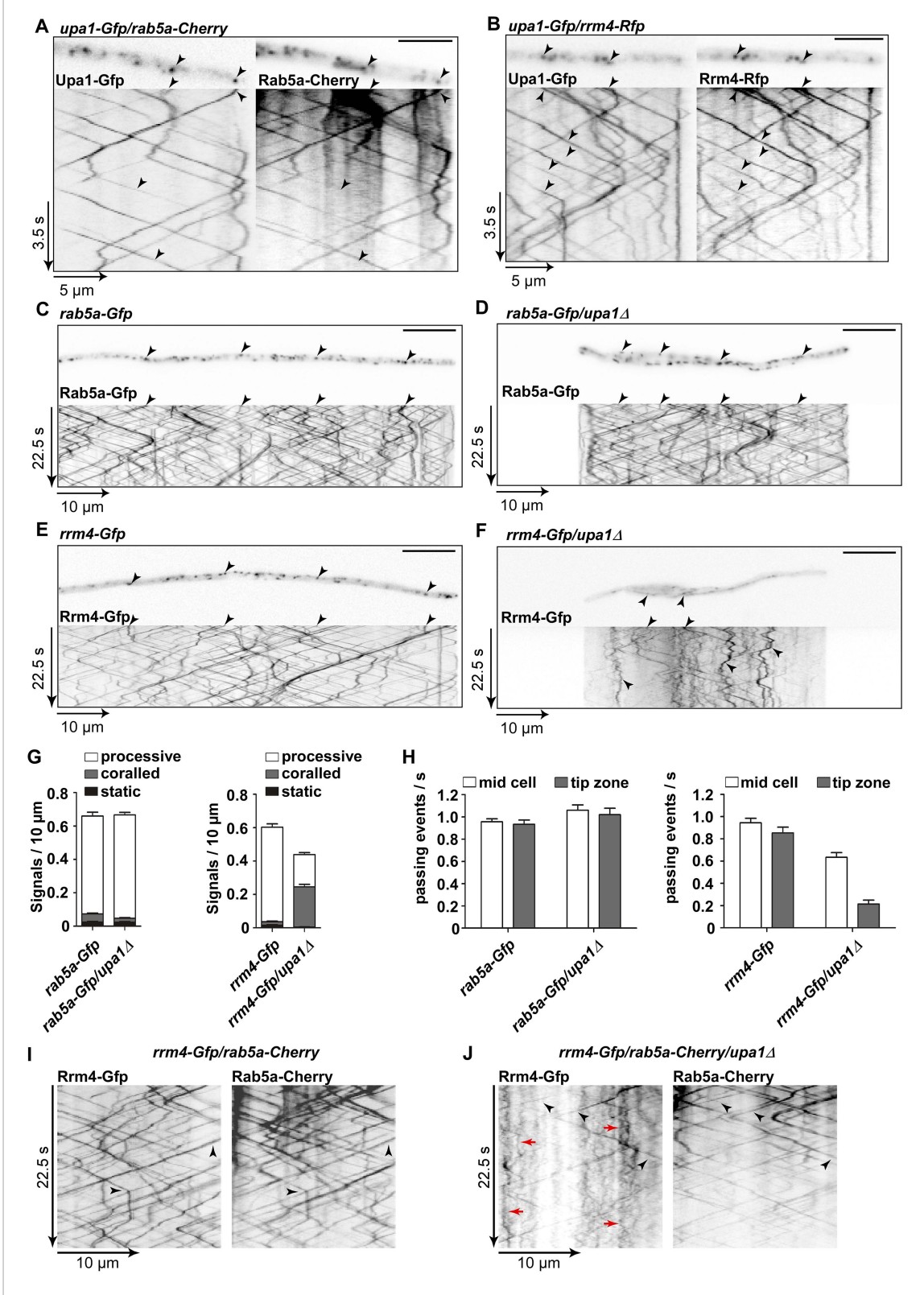

**Figure 5**. Upa1 is crucial for Rrm4 movement on Rab5a-positive endosomes. (**A**) Dynamic co-localisation studies of Upa1-Gfp (left) and Rab5a-Cherry (right) using dual view and msALEX microscopy (see 'Materials and methods'; arrowheads indicate co-localising signals). Micrographs (size bar, 10 μm) and corresponding kymographs of hyphal tip (*Video 12*). (**B**) Dynamic co-localisation studies of Upa1-Gfp (left) and Rrm4-Rfp (right) as in (**A**) (*Video 13*). (**C–F**) Micrographs (size bar, 10 μm) and corresponding kymographs of hyphae expressing Rab5a-Gfp (**C**), Rab5a-Gfp/*upa1Δ* (**D**), Rrm4-Gfp (**E**), or

*Figure 5. continued on next page*

*Figure 5. Continued*

Rrm4-Gfp/*upa1Δ* (**F**) (*Videos 14–17*). (**G**) Bar diagrams depicting amount of Rab5a-Gfp signals per 10 µm hyphae in *wt* and *upa1Δ* cells (left, error bars, s. d.; >22 hyphae), as well as amount of Rrm4-Gfp signals per 10 µm hyphae in *wt* and *upa1Δ* cells (right, error bars, s.d.; >15 hyphae). (**H**) Number of Rab5a-Gfp (left) and Rrm4-Gfp (right)—signals passing zones in the middle of the hyphae and 10 µm from the apical pole in *wt* and *upa1Δ* cells, respectively (passing events of signals/s, error bars, s.d.; more than 15 hyphae). (**I**) Dynamic co-localisation studies of Rrm4-Gfp (left) and Rab5a-Cherry (right) using dual view and msALEX microscopy (arrowheads indicate co-localising signals). (**J**) Same analysis as in (**I**) using a strain carrying a deletion in *upa1*. Corralled movement of Rrm4 signals not found associated with Rab5a is highlighted by red arrows.

The following figure supplements are available for figure 5:

**Figure supplement 1**. Upa1 co-localises with Rab5a and Rrm4, but loss of Upa1 does not affect long-distance transport of endosomes.

**Figure supplement 2**. Rrm4 and Pab1 movement is altered in the absence of Upa1.

**Figure supplement 3**. Rrm4 does not influence endosomal localisation of Upa1.

**Figure supplement 4**. Residual processive movment of Rrm4-Gfp takes place on endosomes.

endosomes in the yeast form via its FYVE domain. Don1 specifically regulates the small GTPase Cdc42 and its efficient endosomal delivery to the site of septation that is crucial for cytokinesis (*Schink and Bölker, 2009*). Here, we demonstrate that the FYVE domain of Upa1 is necessary to target the protein to the identical endosomal compartment, and that this localisation is essential for Upa1 activity specifically during filamentous growth (*Figure 8*).

In contrast to Pib1p and Rififylin, Upa1 contains the aforementioned PAM2, two PAM2Ls and five ankyrin repeats. The latter is a wide-spread protein–protein interaction motif of about 30 amino acids in length found in the human cytoskeletal protein Ankyrin (*Mosavi et al., 2004*; *Li et al., 2006*). Unfortunately, the ankyrin repeats in Upa1 escaped our analysis because of the instability of protein variants lacking the repeats (*Figure 2—figure supplement 2B*). PAM2 motifs are found in a number of interaction partners of the mammalian protein PABPC1, such as translational initiation factor eRF3, nuclease subunit Pan3 and miRNA regulator GW182 (*Hoshino et al., 1999*; *Uchida et al., 2004*; *Tritschler et al., 2010*; *Zekri et al., 2013*). Structural and functional analysis revealed a conserved core motif (consensus xxLNxxAxEFxP; *Kozlov et al., 2010*), which is inserted in the peptide-binding pocket of the MLLE domain (*Albrecht and Lengauer, 2004*; *Kozlov et al., 2004*; *Jinek et al., 2010*). In Upa1, this motif is necessary and sufficient for interaction with the MLLE domain of Pab1, confirming its bioinformatic identification. However, to our surprise the PAM2 motif is dispensable for Upa1 function, which could be explained with a redundant Pab1 interaction

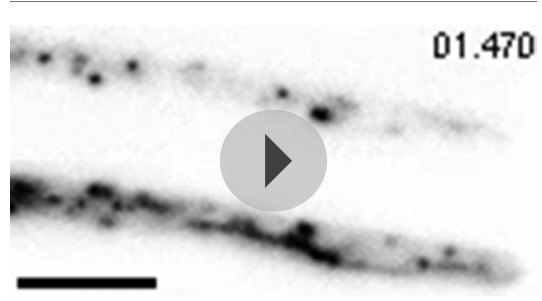

**Video 12.** Upa1-Gfp and Rab5a-Cherry (upper and lower part, respectively) co-localise in shuttling units in hyphae of AB33upa1-Gfp/rab5a-Cherry. Videos were recorded simultaneously using dual-colour detection and correspond to *Figure 5A* (size bar = 5 µm, timescale in seconds, 70 ms alternating exposure time, 200 frames, 15 frames/s display rate; QuickTime format, 561 kB)

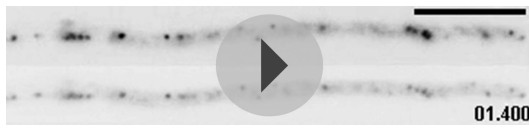

**Video 13.** Upa1-Gfp and Rrm4-Rfp (upper and lower part, respectively) co-localise in shuttling units in hyphae of AB33upa1-Gfp/rrm4-Rfp. Videos were recorded simultaneously using dual-colour detection and correspond to *Figure 5B* (size bar = 10 µm, timescale in seconds, 70 ms alternating exposure time, 100 frames, 15 frames/s display rate; QuickTime format, 132 kB).

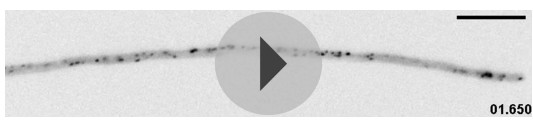

**Video 14.** Rab5a-Gfp moving bidirectionally in a hypha of AB33rab5a-Gfp. Video corresponds to *Figure 5C* (size bar = 10 μm, timescale in seconds, 150 ms exposure time, 150 frames, 6 frames/s display rate; QuickTime format, 248 kB).

**Video 15.** Rab5a-Gfp moving bidirectionally in a hypha of AB33rab5a-Gfp/upa1Δ. Loss of Upa1 results in bipolar growing cell, but does not affect Rab5a-Gfp shuttling. Video corresponds to *Figure 5D* (size bar = 10 μm, timescale in seconds, 150 ms exposure time, 150 frames, 6 frames/s display rate; QuickTime format, 219 kB).

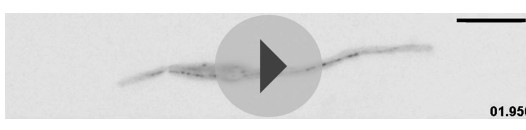

**Video 16.** Rrm4-Gfp moving bidirectionally in a hypha of AB33rrm4-Gfp. Video corresponds to *Figure 5E* (size bar = 10 μm, timescale in seconds, 150 ms exposure time, 150 frames, 6 frames/s display rate; QuickTime format, 2324 kB).

domain. Such a scenario was already described during miRNA regulation in *Drosophila melanogaster*. The key factor GW182 interacts with PABPC1 via two regions, the PAM2 sequence and a second region that provides indirect interaction with PABPC1. Due to these redundant binding modes, the PAM2 motif of GW182 is functionally dispensable for PABP binding and silencing in *D. melanogaster* (*Eulalio et al., 2009*; *Huntzinger et al., 2010*).

Relatedly, we found that Upa1 interacts with MLLE domains of Rrm4 via two PAM2-like sequences. This motif contains a conserved core $^{D}/_{E}$ $^{D}/_{E}$ $^{D}/_{E}$FVYP showing a similarity to the core of the PAM2 sequence (EFxP) including the essential phenylalanine that is inserted into a hydrophobic binding pocket of the MLLE domain (*Kozlov et al., 2004*). Note, that mutations in the phenylalanine of both PAM2-like sequences in Upa1 resulted in loss of Rrm4 interaction (*Figure 3—figure supplement 2*). A phylogenetic sequence comparison revealed that closely related basidiomycetes, which contain an Rrm4 homologue with MLLE domain, also possess Upa1 homologues with at least one of the two PAM2L motifs (*Figure 8—figure supplement 1A*). Experimentally, we found that although both

**Video 17.** Rrm4-Gfp moving bidirectionally in a hypha of AB33rrm4-Gfp/upa1Δ. Loss of Upa1 disturbs shuttling of Rrm4-Gfp, as seen by increased corraled movement. Video corresponds to *Figure 5F* (size bar = 10 μm, timescale in seconds, 150 ms exposure time, 150 frames, 6 frames/s display rate; QuickTime format, 741 kB).

RNA-binding proteins, Pab1 and Rrm4, contain similar MLLE domains (*Becht et al., 2005*, *Figure 8—figure supplement 1B*), they apparently differ in their sequence specificity (*Figure 3F*; *Figure 3—figure supplement 5* and *Figure 3—figure supplement 6*). Thus, we hypothesize that in principle both PAM2 and PAM2L sequences are able to interact with MLLE domains but based on their specific interaction with Rrm4, the novel PAM2L motifs are crucial for endosomal mRNP recruitment (*Figure 8*). In accordance, mutations in the PAM2L motifs lead to loss of Upa1 functionality (*Figure 3H*).

## The endosomal protein Upa1 fulfils specific functions during mRNA and ribosome transport

Based on the fact that Upa1 needs to be present on endosomes and interacts with the RNA-binding proteins Pab1 and Rrm4, we hypothesize that Upa1 specifically functions during endosomal mRNP targeting (*Figure 8*). Importantly, we can exclude that Upa1 is necessary for other known biological functions of this Rab5a-positive endosomal compartment, since septum formation, cytokinesis, FM4-64 uptake, or Rab5a shuttling are not altered in the absence of Upa1 (*Wedlich-Söldner et al., 2002*; *Schink and Bölker, 2009*; *Baumann et al., 2012*). Consistently, also Rrm4 is dispensable for basic functions of endosomes, such as their movement or the association of Rab5a to endosomes (*Baumann et al., 2012*).

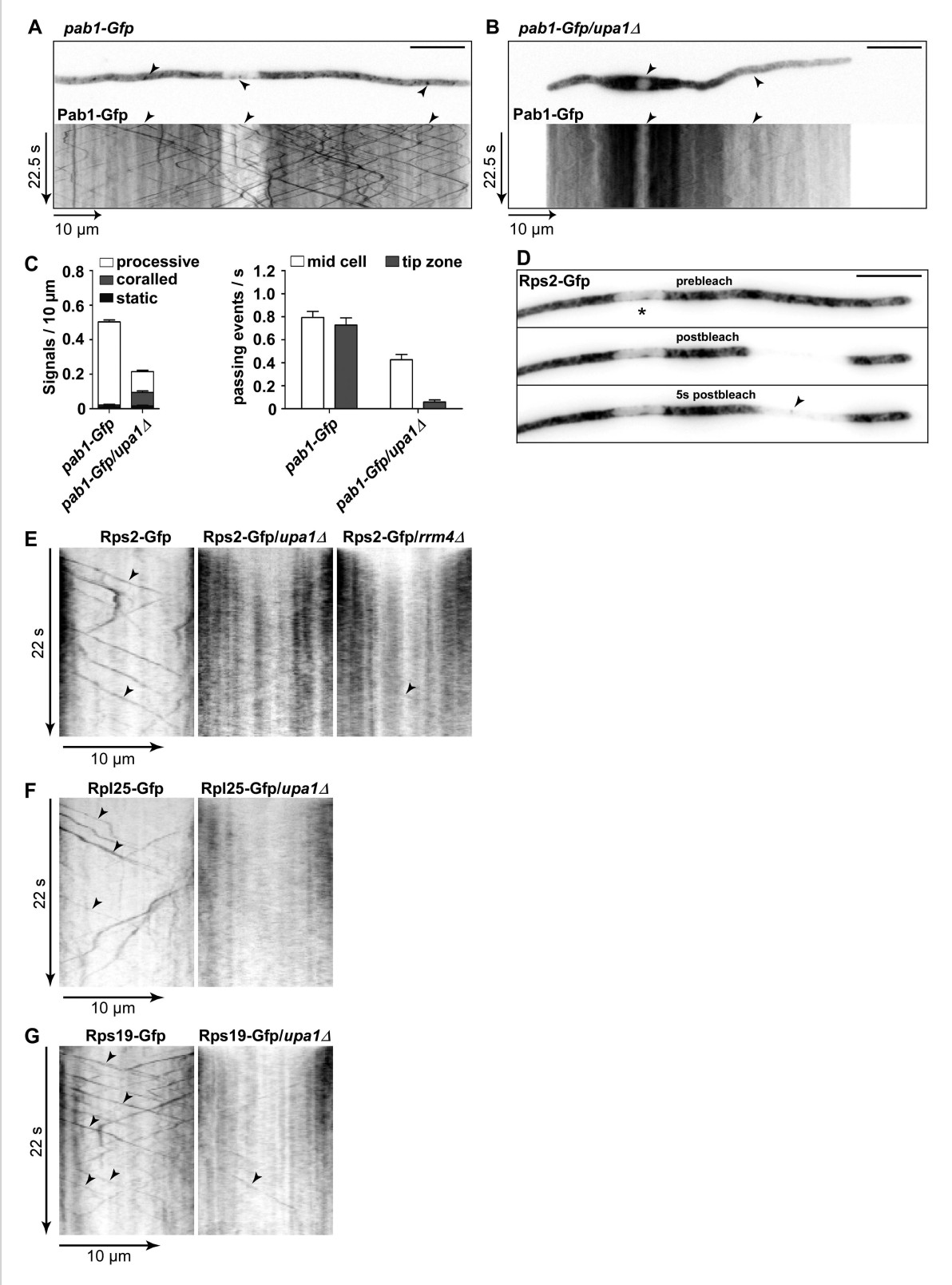

**Figure 6**. Upa1 functions specifically in mRNP function of endosomes. (**A**, **B**) Micrographs (size bar, 10 µm) and corresponding kymographs (*Videos 18*, *19*) of hyphae expressing Pab1-Gfp (**A**) or Pab1-Gfp/*upa1Δ* (**B**). (**C**) Bar diagrams depicting amount of Pab1-Gfp signals per 10 µm hyphae in *wt* and *upa1Δ* cells (left; error bars, s.d.; more than 13 hyphae) and number of Pab1-Gfp signals passing two zones in the middle of the hyphae and 10 µm from the apical pole in *wt* and *upa1Δ* cells (right; passages of signals/s, error bars, s.d.; >13 hyphae). (**D**) Analysing subcellular localisation of ribosomal protein Rps2-Gfp as an example. *Figure 6. continued on next page*

*Figure 6. Continued*

Micrographs (size bar, 10 μm) of hyphae expressing Rps2-Gfp (nucleus indicated by asterisk). 20-μm area was bleached by laser irradiation about 10 μm from the hyphal tip. Arrowhead indicates processive signal entering the bleached area. (**E**, **F**, **G**) Kymographs of hyphal areas bleached with laser irradiation, as shown in (**D**). Arrowheads indicate processive signals.

Importantly, we observed specific defects in Rrm4-dependent endosomal mRNA transport in the absence of Upa1: (i) the amount of aberrant bipolar growing hyphae is increased; (ii) *cts1* secretion is specifically disturbed in the hyphal form; (iii) processive movement of Rrm4 is disturbed; (iv) endosomal transport of the mRNA indicator Pab1, as well as of mRNA-associated ribosomes is strongly reduced; (v) septin mRNA and protein are strongly affected in endosomal transport; (vi) efficient delivery of septin protein to the hyphal tip is disturbed. These observations all point toward the conclusion that Upa1 has a specific function in endosomal targeting of Rrm4 (*Figure 8*). In the absence of Upa1, endosomal Rrm4 functions are disturbed, and consequently, mRNA and associated ribosomes are transported less efficiently. In line with this, specific functions of Rrm4 in septin mRNA and protein transport are also affected, overall leading to characteristic defects of hyphal functions, such as unipolar growth and Cts1 secretion. Thus, Upa1 is a key factor in endosomal recruitment of Rrm4. However, since endosomal movement of Rrm4 was not completely abolished in the absence of Upa1, we envision the presence of additional factors involved (*Figure 8*). Noteworthy, the observed defects in Rrm4 function are fully consistent with our earlier model that Rrm4 mediates local translation of transported mRNAs for the endosomal transport of the translation products (*Figure 8*; *Baumann et al., 2014*; *Jansen et al., 2014*).

## Linking mRNPs to trafficking membranes

Recent data suggest that there is a close connection between mRNA and membrane trafficking (*Kraut-Cohen and Gerst, 2010*; *Jansen et al., 2014*; *Berkovits and Mayr 2015*). Prominent examples are actin-dependent co-transport of mRNAs and ER during budding in *S. cerevisiae* (*Schmid et al., 2006*), endosomal transport of viral RNA (*Ghoujal et al., 2012*) or endosomal miRNA-dependent processes (*Kim et al., 2014*). In this respect, one of the key questions is how mRNPs are connected to membranes. Mammalian p180, for example, contains a lysine-rich RNA-binding domain in concert with a membrane spanning domain and thereby attaches mRNAs to the surface of ER membranes to support their local translation (*Cui et al., 2012*). She2p from yeast harbours a lipid-binding domain and is able to recognise membrane curvature supporting specific ER association of mRNAs during their transport to daughter cells (*Genz et al., 2013*). Furthermore, neuronal PICK1 contains a banana-shaped BAR (Bin-Amphiphysin-Rvs) domain for interaction with curved membranes. Recently, it was shown that this endosome-associated factor specifically interacts with Argonaute 2, a core component of the miRNA machinery. This provides a mechanism of how miRNAs can be attached to endosomes to carry out specific functions, such as miRNA assembly or translational regulation in neurons (*Antoniou et al., 2014*).

Here, we demonstrate that a FYVE domain protein directly couples the key RNA-binding protein of mRNA transport to endosomes by novel PAM2L motifs. Thereby, mRNPs and associated ribosomes are attached to endosomes during microtubule-dependent trafficking. This transport process is important in distributing mRNAs and ribosomes throughout the highly polarised cells (*König et al., 2009*; *Baumann et al., 2012*; *Higuchi et al., 2014*), as well as in the delivery of translation products such as septins to the growing tip (*Baumann et al., 2014*; *Jansen et al., 2014*). In essence, we provide first mechanistic insights into how mRNPs and associated ribosomes are attached to endosomes during long-distance transport.

## Materials and methods

### Standard molecular biology techniques and strain generation

*E. coli* K-12 derivates DH5α (Bethesda Research Laboratories) and Top10 (Life Technologies, Carlsbad, CA, USA) were used for cloning purposes. *S. cerevisiae* strain AH109 (Clontech

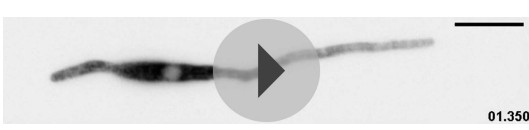

**Video 18.** Pab1-Gfp moving bidirectionally in a hypha of AB33pab1-Gfp. Video corresponds to *Figure 6A* (size bar = 10 μm, timescale in seconds, 150 ms exposure time, 150 frames, 6 frames/s display rate; QuickTime format, 729 kB).

**Video 19.** Pab1-Gfp moving bidirectionally in a hypha of AB33pab1-Gfp/upa1Δ. Loss of Upa1 disturbs shuttling of Pab1-Gfp, as seen by the drastically decreased number of processive Pab1-Gfp signals. Video corresponds to *Figure 6B* (size bar = 10 μm, timescale in seconds, 150 ms exposure time, 150 frames, 6 frames/s display rate; QuickTime format, 353 kB).

Laboratories Inc., Mountain View, CA, USA) was used for yeast two-hybrid analyses. Transformation and cultivation were performed using standard techniques. Growth conditions for *U. maydis* strains and source of antibiotics were described elsewhere (*Brachmann et al., 2004*). Strains were constructed by the transformation of progenitor strains with linearized plasmids. All homologous integration events were verified by Southern blot analysis (*Brachmann et al., 2004*). For ectopic integration, plasmids were linearized with SspI and targeted to the $ip^S$ locus (*Loubradou et al., 2001*). Genomic DNA of wild-type strain UM521 (a1b1) was used as a template for PCR amplifications unless otherwise noted. Detailed information is given in *Supplementary files 2–7*. Accession numbers of *U. maydis* genes used in this study: *upa1* (*UMAG_12183*), *rrm4* (*UMAG_10836*), *pab1* (*UMAG_03494*), *kin3* (*UMAG_06251*), *rab5a* (*UMAG_10615*), *yup1* (*UMAG_05406*), *rps2* (*UMAG_05139*), *rpl25* (*UMAG_05998*), *rps19* (*UMAG_11551*), and *cdc3* (*UMAG_10503*).

## Filamentous growth on solid media

Cell suspensions were grown for about 12 hr in 3 ml CM supplemented with 1% glucose (glc) at 28°C. 4 μl of the densely grown cells were spotted on NM-glc plates containing 1% (wt/vol) activated charcoal, sealed with parafilm and incubated at 28°C for 48hr. Pictures were taken using a Stemi 2000C stereomicroscope (Zeiss, Oberkochen, Germany) with a mounted Canon PowerShot A650 IS camera (Canon Germany GmbH, Krefeld, Germany).

## Fluorometric measurement of endochitinolytic activity

*U. maydis* cell suspensions were grown at 28°C for about 12 hr in 20 ml CM supplemented with 1% glucose (glc) and set to an $OD_{600}$ of 0.5. Either sporidial cells were directly measured or filamentous growth was induced by shifting to NM (1% glc) and subsequent incubation at 28°C for 6–9 hr. 30 μl of the culture was mixed with 70 μl 0.25 μM 4-Methylumbelliferyl β-D-N,N′,N″-triacetylchitotrioside (Sigma–Aldrich, Taufkirchen, Germany), a specific substrate for endochitinolytic activity. After incubation for 1 hr (protected from light), the reaction was stopped by adding 200 μl 1M $Na_2CO_3$. Enzymatic activity was measured by detecting the fluorescent product with a fluorescence spectrometer Infinite M200 (Tecan Group Ltd., Männedorf, Switzerland) using an excitation and emission wavelength of 360 nm and 450 nm, respectively. At least three independent biological experiments were performed with three technical replicates per strain (*Koepke et al., 2011*).

### Yeast two-hybrid analysis

The two-hybrid system Matchmaker 3 from Clontech was used. Yeast two-hybrid strains were co-transformed with derivates of pGBKT7-DS and pGADT7-Sfi (*Supplementary file 5*) and were grown on SD plates without leucine and tryptophan at 28°C for 4 days. Transformants were patched on SD plates without leucine and tryptophan (control) or on SD plates without leucine, tryptophan, histidine, and adenine (selection). Plates were incubated at 28°C for 3 days to test for growth under selection condition. For qualitative plate assays, cells (SD -leu, -trp, $OD_{600}$ of 0.5) were diluted with sterile water in 1:5 steps, and 4 μl drops were spotted on control and selection plates and incubated at 28°C for 3 days. Colony growth was documented with a LAS 4000 imaging system (GE Healthcare Life Sciences, Little Chalfont, United Kingdom). Expression of hybrid proteins was analysed by Western blot (see below).

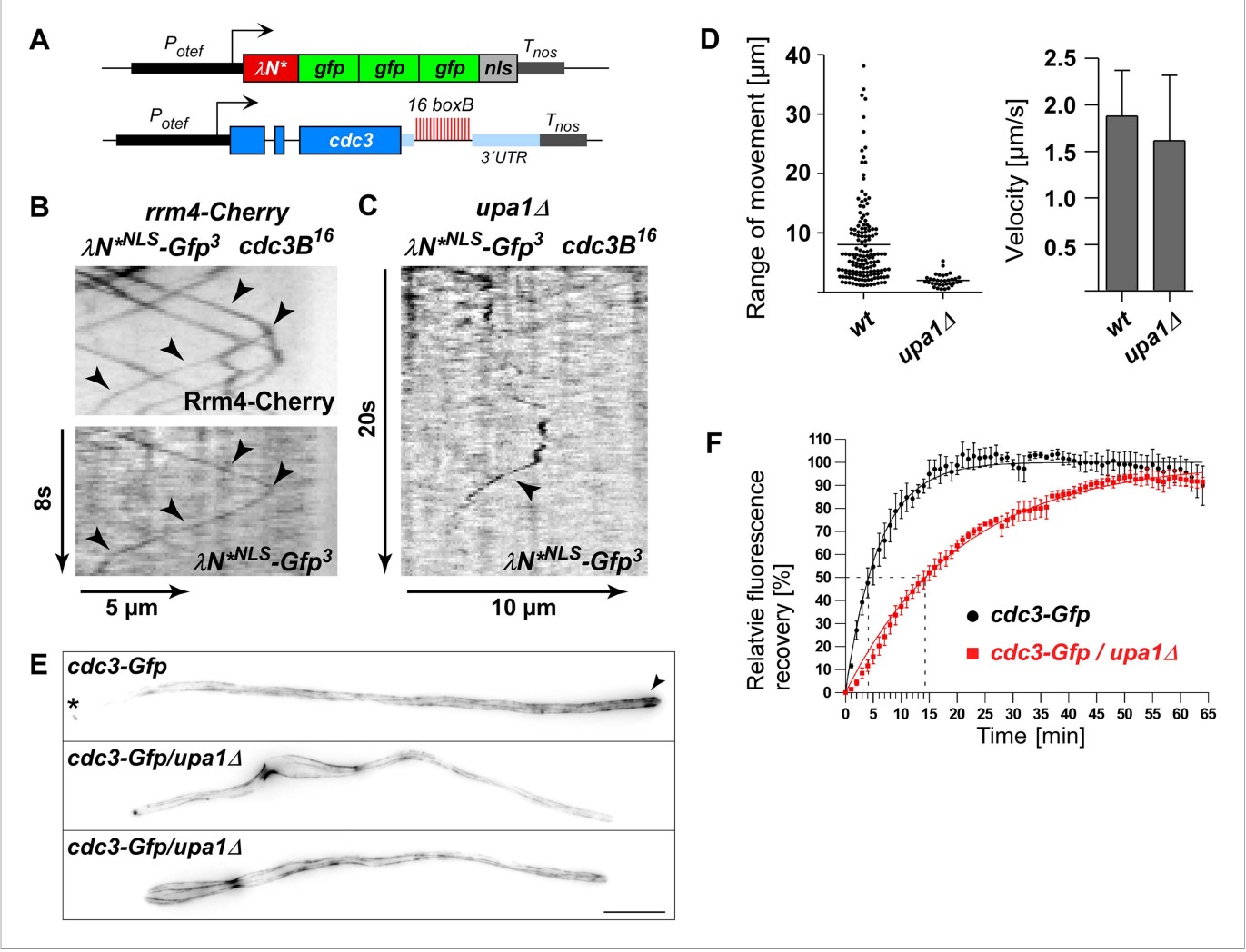

**Figure 7.** Loss of Upa1 disturbs Rrm4-dependent transport of *cdc3* septin mRNA and protein. (**A**) Schematic representation of components of the modified λN RNA reporter system (P_otef, constitutively active promoter; T_nos, heterologous transcriptional terminator; *cdc3B16* carries 16 copies of *boxB* hairpin in its 3' UTR; λN*NLS-Gfp3, modified λN peptide fused to triple Gfp; and NLS; *Baumann et al., 2014*). (**B**) Dynamic co-localisation of strain expressing Rrm4-Cherry, λN*NLS-Gfp3, and *cdc3B16*. Kymograph with directed particles (arrowheads). (**C**) Kymograph of *upa1Δ* strain expressing λN*NLS-Gfp3 protein and *cdc3B16* mRNA. Occasionally directed particles are observed that exhibit altered processive movement. (**D**) Diagram showing range of movement of λN*NLS-Gfp3-labelled mRNAs (vertical bar = mean, 151 mRNA particles for wt and 33 particles for *upa1Δ*) on the left and velocity λN*NLS-Gfp3-labelled mRNAs on the right (error bars, SD; 54 and 96 hyphae for wt and *upa1Δ*, respectively). (**E**) Micrographs of Cdc3G or Cdc3G/*upa1Δ* expressing hyphae (maximum projection of z-stacks with 0.27 µm steps; size bar, 10 µm). Arrowhead marks gradient of septin filaments emanating from the hyphal tip. (**F**) FRAP analysis of Cdc3-Gfp or Cdc3-Gfp/*upa1Δ* expressing hyphae 7–10 h.p.i. (about 10 µm from the hyphal tip; data were fitted to uniphasic exponential equation, dashed lines indicate half time of recovery; n = 3 independent experiments with 4–6 hyphae per experiment; error bars represent s.e.m.). Fluorescence is normalised to plateau (*Baumann et al., 2014*).

The following figure supplement is available for figure 7:

**Figure supplement 1.** Endosome-dependent movement of *cdc3* mRNA and protein.

## Protein extracts and Western blot analysis

Preparation of protein extracts of *U. maydis* and *S. cerevisiae* cells was carried out according to published protocols (*Baumann et al., 2012*; Clontech). For the latter, cells were grown over night at 28°C in SD -leu-trp medium to an $OD_{600}$ of about 0.75. The exact $OD_{600}$ was recorded and together with the culture volume used to calculate the OD-units (50 ml × $OD_{600}$ of 0.75 = 37.5 OD units). Cells

**Video 20.** Video of cdc3B[16] particle visualised by λN*[NLS]-Gfp[3]. Arrowhead at the beginning of the video marks the starting point of the moving particle (size bar = 5 µm, timescale in seconds, 150 ms exposure time, 150 frames, 6 frames/s display rate, QuickTime format, 5495 kB). DOI: 10.7554/eLife.06041.038

were harvested by centrifugation ($2000 \times g$, 5 min) and resuspended in 100 µl yeast cracking buffer (40 mM Tris–HCl [pH 6.8], 8 M Urea, 5% [wt/vol] SDS, 0.1 mM $Na_2$-EDTA, 0.4 mg/ml bromophenol blue, 0.1% [vol/vol] β-mercaptoethanol, 7% [vol/vol] benzamidine, and 5% [vol/vol] PMSF) per 7.5 OD units (e.g., 37.5 OD-units/7.5 = 500 µl YCB). The suspension was transferred to 2 ml reagent tubes, 100 µL of glass beads added, and the sample boiled at 99°C, while shaking (1000 rpm). Immediately, the cells were cooled on ice and subsequently analysed by Western Blot or stored at −70°C. For Western blotting protein samples were resolved by 8% SDS-PAGE and transferred to a PVDF membrane (GE Healthcare) by semi-dry blotting. Western blot analysis was conferred with anti-GFP (clones 7.1 and 13.1), anti-c-Myc (clone 9E10; Roche), anti-alpha-Tubulin (clone DM1A), anti-HA (clone 12CA5) and anti-GST (Sigma) antibodies. A mouse IgG HRP conjugate (H+L; Promega, Madison, WI) was used as a secondary antibody. Activity was detected using the AceGlow blotting detection system (Peqlab, Erlangen, Germany).

## Linker scanning mutagenesis

In order to identify crucial amino acids for the interaction of Upa1 with Rrm4, we performed a linker scanning mutagenesis of plasmid pGBKT7-Upa1$^{\Delta N7/\Delta FR}$-Gfp (aa 883 to 1947) resulting in 10 amino acid substitutions of the sequence AASAAATAAS. Serines and threonine were introduced by the Golden Gate cloning system (*Terfrüchte et al., 2014*) to prevent potential translational problems resulting from 10 consecutive alanines. pGBKT7-Upa1$^{\Delta N7/\Delta FR}$-Gfp was used as a template for two PCR reactions amplifying two products, which lie directly upstream and downstream of the targeted sequence of 30 nucleotides. Oligonucleotide combinations u2 and p2 were used for the upstream situated sequence and combinations d2 and p1 for the downstream sequences (*Supplementary file 7*). The resulting products were subcloned in pDest (pUMa1467, *Terfrüchte et al., 2014*) using BsaI. From these storage plasmids, the mutagenized alleles were introduced into pGBKT7-Upa1$^{\Delta N7\Delta FR}$Gfp as a SfiI/SfiI-fragment. Mutagenesis was verified by an introduced SacII-restriction site as well as by sequencing.

## GST pull downs

Derivates of plasmids pGEX and pET15B (*Supplementary file 6*) were transformed into *E. coli* Rosetta. Overnight cultures were diluted 1:50 in a final volume of 100 ml. Protein expression was induced with IPTG for 4 hr. Cells were pelleted, resuspended in 10 ml lysis buffer (20 mM Tris-Cl, pH 7.5; 200 mM NaCl; 1 mM EDTA; pH 8.0; 0.5% Nonidet P-40; 1 tablet protease inhibitor per 50 ml; Roche, Mannheim, Germany) and lysed by sonication. 50 µl glutathione beads (GE Healthcare) were washed 3 times with lysis buffer. For each pulldown, 500 µl cell lysate with GST-tagged protein was added to the washed beads, incubated for 2 hr at 4°C and subsequently washed 5× with lysis buffer. 1 ml cell lysate containing different Upa1 variants was added directly to loaded GST columns, incubated for 1 hr at 4°C and subsequently washed 5 times with lysis buffer. Beads were boiled 6 min at 99°C. 10 µl of each fraction was loaded on SDS-PAGE for analysis and stained with Coomassie blue.

For protein purification, GST-purification was performed as described above. For GST-tagged proteins, 1.5 ml glutathione beads (GE Healthcare) were equilibrated with lysis buffer. Cell lysate was loaded onto the columns, incubated, and washed. The GST-tagged proteins were eluted in elution buffer (50 mM Tris-HCl; pH 7.5; 200 mM NaCl; 20 mM Glutathione) and glutathione was removed via PD-10 Desalting columns (GE Healthcare). For the His$_6$-tagged Upa1 variants, cells were pelleted, resuspended in His-lysis buffer (50 mM $NaH_2PO_4$; 300 mM NaCl; 10 mM imidazole; pH 8.0) and lysed by sonication. Cell lysates were loaded onto 1.5 ml Ni-NTA agarose columns (Qiagen, Hilden, Germany) and incubated for 1 hr at 4°C. The Ni-NTA agarose was washed 3× with His-lysis buffer (increasing Imidazole concentration from 10 to 20 mM). His-tagged proteins were eluted in His-lysis buffer (containing 250 mM imidazole).

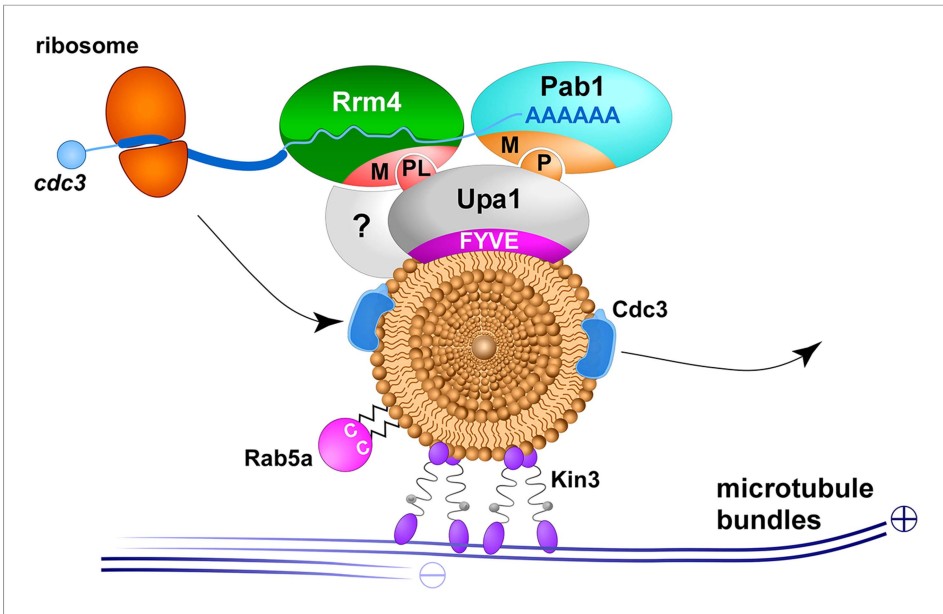

**Figure 8**. Upa1 functions specifically in endosomal mRNA transport. Model proposing Upa1 function during endosomal mRNA transport. Microtubules are given in blue, kinesin-3 type motor Kin3 transports endosomes in the plus-end direction. The small GTPase Rab5a (magenta) marks this specific endosomal compartment, described as early endosomes (*Higuchi et al., 2014*). Upa1 binds endosomes via FYVE domain and the MLLE domain (M) of Pab1 via PAM2 motif (P), as well as the MLLE domain of Rrm4 via PAM2L motif (PL; for simplicity only one motif is shown). Note that the interaction of Upa1 with Pab1 is dispensable, whereas the interaction of Upa1 with Rrm4 is crucial for the endosomal localisation of septin mRNA (blue line with poly[A]-tail), septin protein (blue) and ribosomes (orange). A currently unknown adaptor protein is highlighted with a question mark.
The following figure supplement is available for figure 8:

**Figure supplement 1**. Rrm4 and Upa1 homologues in fungi.

The pulldown experiments with purified proteins were performed similar as described above. Glutathione beads were loaded with100 µl purified GST-tagged proteins and 400 µl lysis buffer. Loaded GST-columns were incubated with 200 µl purified $His_6$-tagged Upa1 variants and 300 µl lysis buffer. Proteins were eluted by boiling, and 10 µl of each fraction was analysed after SDS-PAGE by colloidal Coomassie staining and Western blotting. For the Western blots, membranes were probed with αHis antibody (1/10,000; Sigma) and αGST antibody (1/5,000; Molecular Probes).

## Microscopy, dual-colour imaging, image processing, quantification, and FM4-64 staining

Standard microscopy was carried out with our set-up as described before (*Baumann et al., 2014*). 20 ml cultures of cells were grown to an $OD_{600}$ of 0.5 in CM supplemented with 1% glucose (glc) and shifted to NM (1% glc) to induce filamentous growth for 6–9 hr. For quantification of bipolarity, hyphae were observed with a 63× Plan-Apochromat objective in combination with a Spot Pursuit CCD camera. Pictures of more than 100 cells were taken and scored for unipolar or bipolar growth, as well as for septum formation. At least three independent experiments were performed.

Staining of hyphae with FM4-64 was done as described elsewhere (*Baumann et al., 2012*). Briefly, 500 µl of filament suspension were labelled in 0.8 µM FM4-64 (Life Technologies). After 30–60 s of incubation at room temperature, samples were subjected to microscopic analysis.

For analysis of signal number, velocity and passages through a defined zone of Gfp fusion protein hyphae were observed with a 63× Plan-Apochromat (NA 1.4) in combination with a Spot Pursuit CCD camera. Videos were recorded with an exposure time of 200 ms and 150 frames taken. Kymographs

were generated from these videos and analysed using Metamorph (Version 7.7.0.0; Molecular Devices, Seattle, IL, USA). Signals were counted manually discriminating between processive movement, corralled movement (covered distance of hypha per 22.5 s < 5 µm; rapid changes of direction), or static signals (no movement, straight line). Velocity was determined by quantifying processive signals (movement >5 µm). Note that one signal could exhibit different speeds (i.e., upon reversal of direction). Those velocities were handled as individual data points and not averaged. Passing events were quantified at two defined regions: 10 µm from the apical tip or in the middle of the filament. The number of passing events reflects the overall crossing of signals through this zone. All parts of the microscope systems were controlled by the software package MetaMorph (version 7; Molecular Devices), which was also used for image processing including the adjustment of brightness and contrast, as well as measurements, quantifications, kymographs, and maximum projections of z-stacks. Fluorescence micrographs are displayed inverted unless otherwise stated.

### RNA live imaging

For RNA live imaging, we improved our $\lambda$N-based system (*König et al., 2009*; *Baumann et al., 2014*, *2015*) and fused protein $\lambda$N*Gfp$^3$ to a NLS resulting in $\lambda$N$^{NLS}$Gfp$^3$. Free $\lambda$N$^{NLS}$-Gfp$^3$, which does not bind cytoplasmic $cdc3B^{16}$ mRNA, is targeted to the nucleus, thereby improving the signal to noise ratio in the cytoplasm substantially.

20 ml culture was grown to an $OD_{600}$ of 0.5 in CM supplemented with 1% glucose (glc) and shifted to NM (1% glc) to induce filamentous growth for 9 hr. For excitation of Gfp, the 488 nm laser line was set to 60%. Hyphae were observed with a 63× Plan-Apochromat (NA 1.4) in combination with a CoolSNAP HQ2 camera. Each video was recorded with 150 ms/frame and contained 150 frames.

For quantification of directed movement, kymographs were generated to study the number, velocity, range, and direction of particle movement. For analysis of directionality, particles that reversed direction were counted twice. To determine the average number of particles per 100 µm of hyphae, the total length of hyphae was measured and divided by the number of particles.

For AB33$\lambda$N$^{NLS}$-Gfp$^3$/P$_{otef}$cdc3B$^{16}$, we counted 151 signals in 5024 µm corresponding to about 54 hyphae with an average length of 94 µm. In AB33$\lambda$N$^{NLS}$-Gfp$^3$/P$_{otef}$cdc3B$^{16}$/upa1$\Delta$ 33, particles were detected in 6610 µm corresponding to 96 hyphae with an average length of 69 µm.

### Photobleaching

Photobleaching was adapted from our previous publication (*Baumann et al., 2015*). In order to visualise moving ribosomal proteins, 15 µm of the respective hyphae were photobleached prior to detection of Gfp fluorescence. The 405 nm laser was set to 29% output power. Total bleach time was 77 ms. The 488 nm laser was set to 50% output (exposure time 150 ms, binning 2).

### FRAP

Design and analysis of FRAP experiments was previously described (*Baumann et al., 2014*, *2015*). An area of 16 µm from hyphal tips was bleached with 8.3% laser power. The beam diameter was set to13 pixels, and the bleach time was 7 ms per pixel. Bleaching was carried out in 11 z-planes through fungal hyphae with a z-distance of 0.5 µm. Fluorescence recovery was acquired with an exposure time of 500 ms in a z-stack of 11 planes with a z-distance of 0.5 µm (open camera shutter). Every minute, a z-stack was collected for a period of 65 min.

## Acknowledgements

We acknowledge Dr K Zarnack as well as lab members for valuable discussion and critical reading of the manuscript. We are grateful to U Gengenbacher and S Esch for excellent technical assistance and to Dr R Kahmann from the Max Planck Institute for Terrestrial Microbiology in Marburg for generous support. The work was financed by a grant from the Deutsche Forschungsgemeinschaft to MF (FE 448/5-2) as part of the German/Mexican research group DFG/CONACYT FOR1334 as well as by CEPLAS EXC 1028.

## Additional information

### Funding

| Funder | Grant reference | Author |
|---|---|---|
| Deutsche Forschungsgemeinschaft (DFG) | DFG FOR1334 | Thomas Pohlmann |
| Heinrich-Heine-Universität Düsseldorf | iGRAD MOI | Carl Haag |
| Deutsche Forschungsgemeinschaft (DFG) | EXC 1028 | Michael Feldbrügge |
| Max-Planck-Gesellschaft | | Mario Albrecht |

The funders had no role in study design, data collection and interpretation, or the decision to submit the work for publication.

### Author contributions

TP, Conception and design, Acquisition of data, Analysis and interpretation of data, Drafting or revising the article; SB, Conception and design, Acquisition of data, Analysis and interpretation of data; CH, MA, Acquisition of data, Analysis and interpretation of data; MF, Conception and design, Analysis and interpretation of data, Drafting or revising the article

## Additional files

### Supplementary files

• Supplementary file 1. Potential PAM2-containing proteins from *U. maydis*.

• Supplementary file 2. Description of *U. maydis* strains used in this study.

• Supplementary file 3. Generation of *U. maydis* strains used in this study.

• Supplementary file 4. Description of plasmids used for *U. maydis* strain generation.

• Supplementary file 5. Description of plasmids used for yeast two-hybrid analyses.

• Supplementary file 6. Description of plasmids used for pulldown experiments.

• Supplementary file 7. DNA oligonucleotides used in this study.

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
