## [Decision Letter]

Thank you for sending your work entitled “A FYVE domain protein specifically links mRNA transport to endosome trafficking” for consideration at *eLife*. Your article has been favorably evaluated by Randy Schekman (Senior editor) and three reviewers, one of whom is a member of our Board of Reviewing Editors.

The Reviewing editor and the other reviewers discussed their comments before we reached this decision, and the Reviewing editor has assembled the following comments to help you prepare a revised submission.

This is a thoroughly executed study, which provides an interesting example of a molecular mechanism for linking RNA transport to endosomes. However, a number of points require additional experimental support and/or clarifications.

1) The model in Figure 9 suggests that in the absence of Upa1, Rrm4 and Pab1 will not be connected to endosomes. However, residual movements of Rrm4 and Pab1, albeit altered ones, are still observed. There does not seem to be any information on the colocalisation of Rrm4 and Pab1 with Rab5a in the absence of Upa1. Is the colocalisation lost? Do the residual movements of these proteins still occur on endosomes or on some other organelles? It would be nice if the authors provided some explanation of the corralled movement phenotype. Especially in the case of Rrm4, the brightness of the signals undergoing corralled back-and-forth movements remains constant. Since these are presumably not single Rrm4 molecules, this type of movement can probably not be explained by enhanced release and occasional re-binding of Rrm4 to endosomes.

2) The observation that deletion of Upa1 has a more dramatic effect on Pab1 and ribosomal proteins than on Rrm4 localization is a bit puzzling. This observation does not directly fit with the proposed model and might require at least additional discussion. Is it possible that Pab1, and not Rrm4, is the major functional interactor of Upa1 in vivo? This could in principle be addressed by mutating the PAM2L sequences in Upa1 and assessing the mutant phenotype. Although not essential, an additional Upa1 mutagenesis experiment to make a protein with all PAM2 and PAM2L sequences mutated could test whether there is some redundancy between the Upa1(PAM2L)-Rrm4 and Upa1(PAM2)-Pab1 interactions in vivo (e.g. in mRNA transport) and strengthen the manuscript.

3) In Figure 7, only 4 particles are analyzed for *upa1∆.* This is insufficient, since in the *wt* about 50% of the 36 particles analyzed move the same distance (panel E). It would be nice to analyze more particles (more hyphae) to be able to conclude with more confidence that there is a strong difference in the behaviour of the motile mRNAs between the genotypes.

4) Figure 8 (Figure 7 in the revised version): How does Cdc3G get to the hyphal tip in *upa1∆*? In what percentage of cells/hyphae did the Cdc3G localize to the tip? Again since this analysis was performed in a deletion strain, it would indicate that there are either alternative pathways for mRNA transport and/or other proteins besides Upa1 are required for the process. These data suggest that the model might need adjustment.

---

## [Author Response]

*1) The model in Figure 9 (Figure 8 in the revised version)suggests that in the absence of Upa1, Rrm4 and Pab1 will not be connected to endosomes. However, residual movements of Rrm4 and Pab1, albeit altered ones, are still observed. There does not seem to be any information on the colocalisation of Rrm4 and Pab1 with Rab5a in the absence of Upa1. Is the colocalisation lost? Do the residual movements of these proteins still occur on endosomes or on some other organelles? It would be nice if the authors provided some explanation of the corralled movement phenotype. Especially in the case of Rrm4, the brightness of the signals undergoing corralled back-and-forth movements remains constant. Since these are presumably not single Rrm4 molecules, this type of movement can probably not be explained by enhanced release and occasional re-binding of Rrm4 to endosomes*.

These are very valuable questions that needed to be addressed. To achieve this we focused our analysis on Rrm4 primarily for two reasons. Firstly, the interaction of Rrm4 with Upa1 is functionally significant and therefore more relevant than the interaction of Pab1 with Upa1 (see below). Secondly, in comparison to Rrm4, Pab1 is more difficult to detect mainly because of the extensive cytoplasmic localisation resulting in higher background signals (see Figure 5—figure supplement 2). Hence, we generated strains expressing Rrm4-Gfp with either of the two endosomal markers Rab5a-Cherry or Yup1-Cherry. In both genetic backgrounds we furthermore deleted *upa1* in order to analyse the consequences caused by a loss of Upa1 function. These strains allowed simultaneous detection of both proteins as well as addressing their dynamic movement (Figure 5; Figure 5—figure supplement 4). Moreover, we stained wild-type and *upa1Δ* strains with the lipophilic dye FM4-64 to analyse membrane trafficking (Figure 5—figure supplement 4). We confirm by these co-localisation studies that Rrm4 movement is indeed specifically disturbed in *upa1Δ* strains, while Rab5a and Yup1 movement remains unaffected. With regard to the questions raised we can now conclude the following:

(i) Is the colocalisation lost? The co-localisation is lost for static signals and corralled movement of Rrm4. Also, these signals cannot be stained with FM4-64 suggesting that these are most likely not associated with membrane compartments. We can only speculate that these are either aberrant accumulations of Rrm4-containing mRNPs or that these are Rrm4-containing mRNPs in the process of assembly that cannot be loaded on endosomes, because Upa1 is not supporting their loading. Since at present we cannot differentiate between these possibilities we provided a simple explanation in the text (see below).

(ii) Do the residual movements of these proteins still occur on endosomes or on some other organelles? The residual processive movement of Rrm4 does take place on Rab5a- and Yup1-positive endosomes. To account for the observation that Rrm4 exhibits some residual binding to endosomes in the absence of Upa1 we modified the model (Figure 8). We now include an additional, currently unknown adaptor that interacts with Rrm4 and endosomes.

We incorporated the new data in Figure 5 and Figure 5—figure supplement 4. The text now reads “Importantly, the processive Rrm4-Gfp signals co-localised with Rab5a and the endosomal marker protein Yup1 as well as with the lipophilic dye FM4-64. These co-localisation studies confirm that Rrm4 movement was specifically altered in the absence of Upa1 and that residual processive movement took place on endosomes (Figure 5; Figure 5—figure supplement 4). Rrm4-Gfp signals exhibiting corralled movement did not co-localise with the applied membrane markers suggesting that these large accumulations constitute aberrant forms that fail to associate with the machinery for long-distance transport.”

*2) The observation that deletion of Upa1 has a more dramatic effect on Pab1 and ribosomal proteins than on Rrm4 localization is a bit puzzling. This observation does not directly fit with the proposed model and might require at least additional discussion. Is it possible that Pab1, and not Rrm4, is the major functional interactor of Upa1 in vivo? This could in principle be addressed by mutating the PAM2L sequences in Upa1 and assessing the mutant phenotype. Although not essential, an additional Upa1 mutagenesis experiment to make a protein with all PAM2 and PAM2L sequences mutated could test whether there is some redundancy between the Upa1(PAM2L)-Rrm4 and Upa1(PAM2)-Pab1 interactions in vivo (e.g. in mRNA transport) and strengthen the manuscript*.

There are two possible explanations for the observation that Upa1 has a more dramatic effect on Pab1 and ribosomal proteins than on Rrm4 localisation. Firstly, in principle it is possible that the disturbed interaction of Rrm4 with Upa1 alters the RNA-binding capacity of Rrm4. Therefore, although Rrm4 is still shuttling it may bind less mRNA and hence we observe less Pab1 and less ribosomal proteins. Secondly, the difference might be due to the different detection levels. Rrm4 is very specific for endosomes, in fact in wild-type cells we hardly observed any Rrm4 signal that is not co-localising with endosomes. However, Pab1 and ribosomal proteins are bound to almost all mRNAs in the whole cytoplasm. Therefore, as mentioned above the cytoplasmic signal is higher and endosomal movement more difficult to detect. This is particularly true for ribosomal proteins. Here, we need to bleach an area in the cytoplasm in order to detect ribosomes on shuttling endosomes (Figure 6). Thus, it is more difficult to detect the residual movement of ribosomal proteins and Pab1 in the absence of Upa1.

However, we fully agree that it was a weakness of the initial manuscript that a functional analysis of the PAM2L motifs of Upa1 was not carried out. To overcome this, we generated strains expressing Upa1-Gfp versions with mutations in the PAM2L motifs (Upa1^mPL1^-Gfp and Upa1^mPL2^-Gfp) as well as a strains with mutations in both PAM2L motifs (Upa1^mPL1+2^-Gfp), and with mutations in PAM2 and PAM2L motifs (Upa1^mP/mPL1+2^-Gfp; Figure 3). All Upa1 versions localised to shuttling endosomes with comparable signal intensity indicating their expression at comparable levels (Figure 3—figure supplement 7). Testing unipolar growth (Figure 3) and Cts1 secretion (Figure 3—figure supplement 7) gave consistent results. One PAM2L motif is sufficient for function, but if both PAM2L motifs were mutated Upa1 functionality was lost. The PAM2 motif did not contribute to these defects. Showing that the PAM2L motifs are functionally important and that therefore the interaction of Rrm4 with Upa1 is more relevant than the interaction of Upa1 with Pab1, considerably improved the statement.

*3) In*
Figure 7*, only 4 particles are analyzed for* upa1∆. *This is insufficient, since in the* wt *about 50% of the 36 particles analyzed move the same distance (panel E). It would be nice to analyze more particles (more hyphae) to be able to conclude with more confidence that there is a strong difference in the behaviour of the motile mRNAs between the genotypes*.

We analysed additional hyphae and quantified more particles. Instead of 13 hyphae for wild-type and the *upa1Δ* strain we now analysed 54 and 96, respectively. The number of particles was increased from 36 to 151 for wild type and from 4 to 33 in *upa1Δ* strains. After improving the analysis we are now more confident in concluding that there is a strong difference in mRNA motility (Figure 7).

*4)*
Figure 8*: How does Cdc3G get to the hyphal tip in* upa1∆*? In what percentage of cells/hyphae did the Cdc3G localize to the tip? Again since this analysis was performed in a deletion strain, it would indicate that there are either alternative pathways for mRNA transport and/or other proteins besides Upa1 are required for the process. These data suggest that the model might need adjustment*.

In order to answer these questions we have to recapitulate earlier results that were published in [4] (EMBO Reports). Septin *cdc3* mRNA and encoded protein localise to shuttling Rrm4 and Rab5a-positive endosomes. In addition, the septin Cdc3-Gfp forms filaments with a gradient emanating from the hyphal tip (Figure 7). In the absence of Rrm4, endosomal localisation of septin mRNA and septin protein is severely disturbed. Without Rrm4 and thus without mRNA transport Cdc3-Gfp stills forms filaments that localise to the hyphal tip while the gradient is no longer formed. Thus, without mRNA transport the efficient assembly of Cdc3G in filaments is affected. This is supported by FRAP experiments showing that in the absence of mRNA transport the recovery of Cdc3-Gfp is significantly delayed.

Based on these earlier results and the results presented here we can hypothesize that diffusion is the most likely way in *upa1Δ* strains that Cdc3-Gfp gets to the hyphal tip. We have shown earlier that for free Gfp this process takes place in seconds. Cdc3-Gfp localises to the tip in all hyphae, but the gradient is disturbed in the absence of Upa1 (Figure 7), reminiscent of the altered gradient in *rrm4* deletion strains (4). In essence, regarding the Cdc3 septin biology loss of Upa1 phenocopies loss of Rrm4. Therefore, we believe that Upa1 is a key factor for the recruitment of Rrm4 to endosomes. However, as out-lined above there is still residual shuttling of Rrm4 on endosomes and therefore we improved the model including the presence of an as yet unknown adaptor (Figure 8).